# Dynamic Bottleneck for Robust Self-Supervised Exploration

**Chenjia Bai**
Harbin Institute of Technology
bai_chenjia@stu.hit.edu.cn

**Lingxiao Wang**
Northwestern University
lingxiaowang2022@u.northwestern.edu

**Lei Han**
Tencent Robotics X
lxhan@tencent.com

**Animesh Garg**
University of Toronto, Vector Institute, NVIDIA
garg@cs.toronto.edu

**Jianye Hao**
Tianjin University
jianye.hao@tju.edu.cn

**Peng Liu**
Harbin Institute of Technology
pengliu@hit.edu.cn

**Zhaoran Wang**
Northwestern University
zhaoranwang@gmail.com

## Abstract

Exploration methods based on pseudo-count of transitions or curiosity of dynamics have achieved promising results in solving reinforcement learning with sparse rewards. However, such methods are usually sensitive to environmental dynamics-irrelevant information, e.g., white-noise. To handle such dynamics-irrelevant information, we propose a Dynamic Bottleneck (DB) model, which attains a dynamics-relevant representation based on the information-bottleneck principle. Based on the DB model, we further propose DB-bonus, which encourages the agent to explore state-action pairs with high information gain. We establish theoretical connections between the proposed DB-bonus, the upper confidence bound (UCB) for linear case, and the visiting count for tabular case. We evaluate the proposed method on Atari suits with dynamics-irrelevant noises. Our experiments show that exploration with DB bonus outperforms several state-of-the-art exploration methods in noisy environments.

## 1 Introduction

The tradeoff between exploration and exploitation has long been a major challenge in reinforcement learning (RL) [35, 50, 58]. Generally, excessive exploitation of the experience suffers from the potential risk of being suboptimal, whereas excessive exploration of novel states hinders the improvement of the policy. A straightforward way to tackle the exploration-exploitation dilemma is to enhance exploration efficiency while keeping exploitation in pace. When the extrinsic rewards are dense, reward shaping is commonly adopted for efficient exploration. However, in many real-world applications such as autonomous driving [34], the extrinsic rewards are sparse, making efficient exploration a challenging task in developing practical RL algorithms. The situations become even worse when the extrinsic rewards are entirely unavailable. In such a scenario, the task of collecting informative trajectories from exploration is known as the self-supervised exploration [11].

An effective approach to self-supervised exploration is to design a dense intrinsic reward that motivates the agent to explore novel transitions. Previous attempts include count-based [9] and curiosity-driven [39] explorations. The count-based exploration builds a density model to measure the pseudo-count of state visitation and assign high intrinsic rewards to less frequently visited states. In contrast, the

curiosity-driven methods maintain a predictive model of the transitions and encourage the agent to visit transitions with high prediction errors. However, all these methods becomes unstable when the states are noisy, e.g., containing dynamics-irrelevant information. For example, in autonomous driving tasks, the states captured by the camera may contain irrelevant objects, such as clouds that behave similar to Brownian movement. Hence, if we measure the novelty of states or the curiosity of transitions through raw observed pixels, exploration are likely to be affected by the dynamics of these irrelevant objects.

To encourage the agent to explore the most informative transitions of dynamics, we propose a Dynamic Bottleneck (DB) model, which generates a dynamics-relevant representation $Z_t$ of the current state-action pair $(S_t, A_t)$ through the Information-Bottleneck (IB) principle [55]. The goal of training DB model is to acquire dynamics-relevant information and discard dynamics-irrelevant features simultaneously. To this end, we maximize the mutual-information $I(Z_t; S_{t+1})$ between a latent representation $Z_t$ and the next state $S_{t+1}$ through maximizing its lower bound and using contrastive learning. Meanwhile, we minimize the mutual-information $I([S_t, A_t]; Z_t)$ between the state-action pair and the corresponding representation to compress dynamics-irrelevant information. Based on our proposed DB model, we further construct a DB-bonus for exploration. DB-bonus measures the novelty of state-action pairs by their information gain with respect to the representation computed from the DB model. We show that the DB-bonus are closely related to the provably efficient UCB-bonus in linear Markov Decision Processes (MDPs) [1] and the visiting count in tabular MDPs [3, 22]. We further estimate the DB-bonus by the learned dynamics-relevant representation from the DB model. We highlight that exploration based on DB-bonus directly utilize the information gain of the transitions, which filters out dynamics-irrelevant noise. We conduct experiments on the Atari suit with dynamics-irrelevant noise injected. Results demonstrate that our proposed self-supervised exploration with DB-bonus is robust to dynamics-irrelevant noise and outperforms several state-of-the-art exploration methods.

## 2    Related Work

Our work is closely related to previous exploration algorithms that construct intrinsic rewards to quantify the novelty of states and transitions. Several early approaches directly define the pseudo-count by certain statistics to measure the novelty of states [41, 30]; more recent methods utilize density model [9, 38] or hash map [52, 42] for state statistics. Nevertheless, these approaches are easily affected by dynamics-irrelevant information such as white-noise. The contingency awareness method [16] addresses such an issue by using an attentive model to locate the agent and computes the pseudo-count based on regions around the agent. However, such an approach could ignore features that are distant from the agent but relevant to the transition dynamics. Another line of research measures the novelty through learning a dynamics model and then use the prediction error to generate an intrinsic reward. These methods are known as the curiosity-driven exploration algorithms. Similar to the pseudo-count based methods, curiosity-driven methods become unstable in the presence of noises, because the prediction model is likely to yield high error for stochastic inputs or targets. Some recent attempts improve the curiosity-driven approach by learning the inverse dynamics [39] and variational dynamics [7] to define curiosity, or utilizes the prediction error of a random network to construct intrinsic rewards [12]. However, without explicitly removing dynamics-irrelevant information, these methods are still vulnerable to noises in practice [11].

The entropy-based exploration uses state entropy as the intrinsic reward. VISR [19], APT [29] and APS [28] use unsupervised skill discovery for fast task adaptation. In the unsupervised stage, they use $k$-nearest-neighbor entropy estimator to measure the entropy of state, and then use it as the intrinsic reward. RE3 [47] and ProtoRL [59] use random encoder and prototypes to learn the representation and use state-entropy as bonuses in exploration. Nevertheless, the state entropy will increase significantly if we inject noises in the state space. The entropy-based exploration will be misled by the noises. Previous approaches also quantify the epistemic uncertainty of dynamics through Bayesian network [21], bootstrapped $Q$-functions [37, 8], ensemble dynamics [40], and Stein variational inference [43] to tackle noisy environments. However, they typically require either complicated optimization methods or large networks. In contrast, DB learns a dynamics-relevant representation and encourages exploration by directly accessing the information gain of new transitions via DB-bonus.

Another closely related line of studies uses the mutual information to promote exploration in RL. Novelty Search (NS) [53] proposes to learn a representation through IB. Curiosity Bottleneck

(CB) [26] also performs exploration based on IB by measuring the task-relevant novelty. However, both NS and CB require extrinsic rewards to learn a value function and are not applicable for self-supervised exploration. Moreover, NS contains additional $k$-nearest-neighbor to generate intrinsic reward and representation loss to constrain the distance of consecutive states, which are costly for computation. In contrast, our DB model handles self-supervised exploration without accessing extrinsic rewards. EMI [25] learns a representation by maximizing the mutual information in the forward dynamics and the inverse dynamics , which is different from the IB principle used in our method. In addition, we aim to perform robust exploration to overcome the white-noise problem, while EMI does not have an explicit mechanism to address the noise.

Our work is also related to representation learning in RL. DrQ [60], RAD [27] and CURL [49] learn the state representation by data augmentation and contrastive learning [14, 20, 36] to improve the data-efficiency of DRL. Deep InfoMax [33] and Self-Predictive Representation (SPR) [46] learn the contrastive and predictive representations of dynamics, respectively, and utilize such representations as auxiliary losses for policy optimization. However, none of these existing approaches extracts information that benefits exploration. In contrast, we show that the dynamics-relevant representation learned by DB can be utilized for efficient exploration.

## 3  The Dynamic Bottleneck

In this section, we introduce the objective function and architecture of the DB model. We consider an MDP that can be described by a tuple $(\mathcal{O}, \mathcal{A}, \mathbb{P}, r, \gamma)$, which consists of the observation space $\mathcal{O}$, the action space $\mathcal{A}$, the transition dynamics $\mathbb{P}$, the reward function $r$, and the discount factor $\gamma \in (0, 1)$. At each time step, an agent decides to perform an action $a_t \in \mathcal{A}$ after observing $o_t \in \mathcal{O}$, and then the observation transits to $o_{t+1}$ with a reward $r_t$ received. In this paper, we use upper letters, such as $O_t$, to denote random variables and the corresponding lower case letter, such as $o_t$, to represent their corresponding realizations.

We first briefly introduce the IB principle [55]. In supervised setting that aims to learn a representation $Z$ of a given input source $X$ with the target source $Y$, IB maximizes the mutual information between $Z$ and $Y$ (i.e. $\max I(Z; Y)$) and restricts the complexity of $Z$ by using the constrain as $I(Z; X) < I_c$. Combining the two terms, the objective of IB is equal to $\max I(Z; Y) - \alpha I(Z; X)$ with the introduction of a Lagrange multiplier.

DB follows the IB principle [55] to learn dynamics-relevant representation. The input variable of the DB model is a tuple $(O_t, A_t)$ that contains the current observation and action, and the target is the next observation $O_{t+1}$. We denote by $S_t$ and $S_{t+1}$ the encoding of observations $O_t$ and $O_{t+1}$. The goal of the DB model is to obtain a compressed latent representation $Z_t$ of $(S_t, A_t)$, that preserves the information that is relevant to $S_{t+1}$ only. Specifically, we use $f_o^S$ and $f_m^S$ as the encoders of two consecutive observations $o_t$ and $o_{t+1}$, respectively. We parameterize the dynamics-relevant representation $z_t$ by a Gaussian distribution with parameter $\phi$, and it takes $(s_t, a_t)$ as input. We summarize the DB model as follows,

$$s_t = f_o^S(o_t; \theta_o), \quad s_{t+1} = f_m^S(o_{t+1}; \theta_m), \quad z_t \sim g^Z(s_t, a_t; \phi). \tag{1}$$

Following the IB principle, the objective of the DB model seeks to maximize the mutual information $I(Z_t; S_{t+1})$ while minimizing the mutual information $I([S_t, A_t]; Z_t)$. To this end, we propose the DB objective by following the IB Lagrangian [55], which takes the form of

$$\min -I(Z_t; S_{t+1}) + \alpha_1 I([S_t, A_t]; Z_t). \tag{2}$$

Here $\alpha_1$ is a Lagrange multiplier that quantifies the amount of information about the next state preserved in $Z_t$. Fig. 1 illustrates the DB objective. We minimize $I([S_t, A_t]; Z_t)$ and consider it as a regularizer in the representation learning. Then the representation learning is done by maximizing the mutual information $I(Z_t, S_{t+1})$. Maximizing $I(Z_t, S_{t+1})$ ensures that we do not discard useful information from $(S_t, A_t)$. In DB, the mutual information is estimated by several variational bounds parameterized by neural networks to enable differentiable and tractable computations. In what follows, we propose a lower bound of (2), which we optimize to train the DB model.

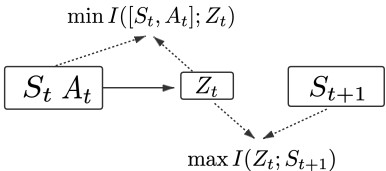

Figure 1: Illustration of DB objective. DB minimizes the mutual information $I([S_t, A_t]; Z_t)$ to obtain a compressive representation, and maximize $I(Z_t; S_{t+1})$ to preserve the information.

## 3.1 Maximizing the lower bound of $I(Z_t; S_{t+1})$

As directly maximizing $I(Z_t; S_{t+1})$ is intractable, we propose to optimize a predictive objective, which is a lower bound of $I(Z_t; S_{t+1})$ [2]. It holds that

$$
\begin{aligned}
I(Z_t; S_{t+1}) &= \mathbb{E}_{p(z_t, s_{t+1})} \left[ \log \frac{p(s_{t+1}|z_t)}{p(s_{t+1})} \right] \\
&= \mathbb{E} \left[ \log \frac{q(s_{t+1}|z_t; \psi)}{p(s_{t+1})} \right] + D_{\mathrm{KL}}[p(s_{t+1}|z_t) \| q(s_{t+1}|z_t; \psi)],
\end{aligned}
\tag{3}
$$

where $p(s_{t+1}|z_t)$ is an intractable conditional distribution and $q(s_{t+1}|z_t; \psi)$ is a tractable variational decoder with parameter $\psi$. By the non-negativity of the KL-divergence, we obtain the following lower bound,

$$
I(Z_t; S_{t+1}) \geq \mathbb{E}_{p(z_t, s_{t+1})}[\log q(s_{t+1}|z_t; \psi)] + \mathcal{H}(S_{t+1}),
$$

where $\mathcal{H}(\cdot)$ is the entropy. Since $\mathcal{H}(S_{t+1})$ is irrelevant to the parameter $\psi$, maximizing $I(Z_t; S_{t+1})$ is equivalent to maximizing the following lower bound,

$$
I_{\mathrm{pred}} \triangleq \mathbb{E}_{p(z_t, s_{t+1})}[\log q(s_{t+1}|z_t; \psi)].
\tag{4}
$$

$I_{\mathrm{pred}}$ can be interpreted as the log-likelihood of the next-state encoding $s_{t+1}$ given the dynamics-relevant representation $z_t$. In practice, we parameterize the prediction head $q(s_{t+1}|z_t; \psi)$ by a neural network that outputs a diagonal Gaussian random variable. Since $s_t$, $s_{t+1}$ and $z_t$ are all low-dimensional vectors instead of raw image pixels, optimizing $I_{\mathrm{pred}}$ is computationally efficient.

**Momentum Encoder**    To encode the consecutive observations $o_t$ and $o_{t+1}$, we adopt the Siamese architecture [10] that uses the same neural network structures for the two encoders. Nevertheless, we observe that if we train both the encoders by directly maximizing $I_{\mathrm{pred}}$, the Siamese architecture tends to converge to a collapsed solution. That is, the generated encodings appears to be uninformative constants. A simple fact is that if both the encoders generate zero vectors, predicting zeros conditioning on $z_t$ (or any variables) is a trivial solution. To address such issue, we update the parameter $\theta_o$ of $f_o^S$ in (1) by directly optimizing $I_{\mathrm{pred}}$. Meanwhile, we update the parameter $\theta_m$ of $f_m^S$ by a momentum moving average of $\theta_o$, which takes the form of $\theta_m \leftarrow \tau \theta_m + (1 - \tau)\theta_o$. In the sequel, we call $f_o^S$ the online encoder and $f_m^S$ the *momentum* encoder, respectively. Similar techniques is also adopted in previous study [18, 46] to avoid the mode collapse.

## 3.2 Contrastive Objective for Maximizing $I(Z_t; S_{t+1})$

In addition to the lower bound of $I(Z_t; S_{t+1})$ in §3.1, we also investigate the approach of maximizing the mutual information by contrastive learning (CL) [36]. CL classifies positive samples and negative samples in the learned representation space. An advantage of adopting CL is that training with negative samples plays the role of regularizer, which avoids collapsed solutions. Moreover, the contrastive objective yields a variational lower bound of the mutual information $I(Z; S_{t+1})$. To see such a fact, note that by the Bayes rule, we have

$$
I(Z_t; S_{t+1}) \geq \mathbb{E}_{p(z_t, s_{t+1})} \mathbb{E}_{S^-} \left[ \log \frac{\exp(h(z_t, s_{t+1}))}{\sum_{s_j \in S^- \cup s_{t+1}} \exp(h(z_t, s_j))} \right] \triangleq I_{\mathrm{nce}}.
\tag{5}
$$

Here $h$ is a score function which assigns high scores to positive pairs and low score to negative pairs. We refer to Appendix A for a detailed proof of (5). The right-hand side of (5) is known as

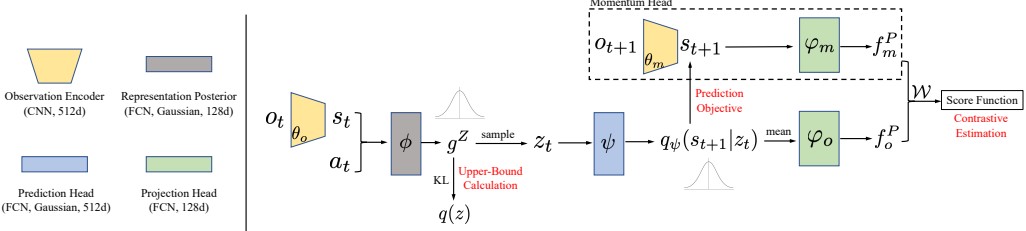

Figure 2: The network architecture of DB. The architecture contains several convolution neural networks (CNNs) and fully-connected networks (FCNs). It consists of four components, including (i) the observation encoder $(f_o^S, f_m^S)$, which consists of an online network $f_o^S$ and a momentum network $f_m^S$; (ii) the representation posterior $g^Z(s_t, a_t; \phi)$, which generates a Gaussian distribution for the input state action pair $(s_t, a_t)$; (iii) the prediction head $q_\psi(s_{t+1}|z_t)$, which predicts the next state $s_{t+1}$ based on sampling representation $z_t$ from the representation posterior; and (iv) the projection heads $(f_o^P, f_m^P)$, which maps the next observation encoding $s_{t+1}$ and its prediction from the prediction head to low-dimensional space to perform contrastive estimation.

the InfoNCE objective [36]. The positive samples are obtained by directly sampling the transitions $(s, a, s')$. In contrast, the negative samples are obtained by first sampling a state-action pair $(s, a)$, and then sampling a state $\tilde{s}$ independently. Then a negative sample is obtained by concatenating them together to form a tuple $(s, a, \tilde{s})$. The negative samples do not follow the transition dynamics. In practice, we collect the negative sample by sampling observation encodings randomly from the batch. We remark that comparing with methods that require data augmentation to construct negative samples [20, 49], DB utilizes a simple scheme to obtain positive and negative samples from on-policy experiences.

In (5), we adopt the standard bilinear function as the score function $h$, which is defined as follows,

$$h(z_t, s_{t+1}) = f_o^P(\bar{q}(z_t; \psi))^\top \mathcal{W} f_m^P(s_{t+1}), \tag{6}$$

where $f_o^P(\cdot; \varphi_o)$ and $f_m^P(\cdot; \varphi_m)$ project $s_{t+1}$ and the mean value of next-state prediction $q(s_{t+1}|z_t; \varphi)$, i.e., $\bar{q}(\cdot; \psi)$, to a latent space to apply the contrastive loss $I_{\text{nce}}$ in (5), and $\mathcal{W}$ is the parameter of the score function. Similar to the observation encoder and MoCo-based architectures [49, 20, 15], we also adopt an online projector $f_o^P$ and a momentum projector $f_m^P$ for $z_t$ and $s_{t+1}$, respectively. The momentum projector is updated by $\varphi_m \leftarrow \tau \varphi_m + (1 - \tau)\varphi_o$.

### 3.3 Minimizing the Upper Bound of $I([S_t, A_t]; Z_t)$

We minimize the mutual information $I([S_t, A_t]; Z_t)$ through minimizing a tractable upper bound of the mutual information. To this end, we introduce a variational approximation $q(z_t)$ to the intractable marginal $p(z_t) = \int p(s_t, a_t)p(z_t|s_t, a_t)ds_t a_t$. Specifically, the following upper-bound of $I([S_t, A_t]; Z_t)$ holds,

$$\begin{aligned} I([S_t, A_t]; Z_t) = \mathbb{E}_{p(s_t, a_t)}\Big[\frac{p(z_t|s_t, a_t)}{p(z_t)}\Big] &= \mathbb{E}_{p(s_t, a_t)}\Big[\frac{p(z_t|s_t, a_t)}{q(z_t)}\Big] - D_{\text{KL}}\big[p(z_t)\|q(z_t)\big] \\ &\leq \mathbb{E}_{p(s_t, a_t)}\big[D_{\text{KL}}[p(z_t|s_t, a_t)\|q(z_t)]\big] \triangleq I_{\text{upper}}, \end{aligned} \tag{7}$$

where the inequality follows from the non-negativity of the KL divergence, and $q(z_t)$ is an approximation of the marginal distribution of $Z_t$. We follow Alemi et al. [2] and use a standard spherical Gaussian distribution $q(z_t) = \mathcal{N}(0, \mathbf{I})$ as the approximation. The expectation of $I_{\text{upper}}$ is estimated by sampling from on-policy experiences.

### 3.4 The Loss Function and Architecture

The final loss for training the DB model is a combination of the upper and lower bounds established in previous sections,

$$\min_{\theta_o, \phi, \psi, \varphi_o, \mathcal{W}} \mathcal{L}_{\text{DB}} = \alpha_1 I_{\text{upper}} - \alpha_2 I_{\text{pred}} - \alpha_3 I_{\text{nce}}, \tag{8}$$

where $\alpha_1$, $\alpha_2$ and $\alpha_3$ are hyper-parameters. As we show in the ablation study (§5), all of the three components in the loss plays an important role in learning dynamics-relevant representations. We illustrate the architecture of the DB model in Fig. 2. In practice, we minimize $\mathcal{L}_{\mathrm{DB}}$ in (8) by gradient descent, which iteratively updates the parameters of $f_o^S, g^Z, q_\psi, \mathcal{W}$ and $f_o^P$. Meanwhile, we adopt exponential moving average to update the parameters of $f_m^S$ and $f_m^P$ to avoid collapsed solutions. We refer to Appendix B for the pseudocode of training DB model.

# 4 Exploration with DB-Bonus

We are now ready to introduce the DB-bonus $r^{\mathrm{db}}$ for exploration. In this section, we first present the DB-bonus for self-supervised exploration. We establish the theoretical connections between the DB-bonus and provably efficient bonus functions. We further present the empirical estimation of DB-bonus and the policy optimization algorithm that utilizes the DB-bonus.

In the sequel, we assume that the learned parameter $\Theta$ of the DB model follows a Bayesian posterior distribution given the training dataset $\mathcal{D}_m = \{(s_t^i, a_t^i, s_{t+1}^i)\}_{i \in [0,m]}$, which is a collection of past experiences from $m$ episodes performed by the agent to train the DB model. We aim to estimate the following conceptual reward, which is defined by the mutual information between the parameter of the DB model and the transition dynamics given the training dataset,

$$
\begin{aligned}
r^{\mathrm{db}}(s_t, a_t) &\triangleq I\big(\Theta; (s_t, a_t, S_{t+1})|\mathcal{D}_m\big)^{1/2} \\
&= \Big[\mathcal{H}\big((s_t, a_t, S_{t+1})|\mathcal{D}_m\big) - \mathcal{H}\big((s_t, a_t, S_{t+1})|\Theta, \mathcal{D}_m\big)\Big]^{1/2}.
\end{aligned}
\tag{9}
$$

Intuitively, DB-bonus defined in (9) encourages the agent to explore transitions that are maximally informative to the improvement of the DB model.

## 4.1 Theoretical Analysis

We show that the DB-bonus defined in (9) enjoys well theoretical properties, and establish theoretical connections between $r^{\mathrm{db}}$ and bonuses based on the optimism in the face of uncertainty [4, 23], which incorporates UCB into value functions in both tabular [5, 22, 17] and linear MDPs [24, 13].

**Connection to UCB-bonus in linear MDPs** In linear MDPs, the transition kernel and reward function are assumed to be linear. In such a setting, LSVI-UCB [24] provably attains a near-optimal worst-case regret, and we refer to Appendix C.1 for the details. The idea of LSVI-UCB is using an optimistic $Q$-value, which is obtained by adding an UCB-bonus $r^{\mathrm{ucb}}$ [1] to the estimation of the $Q$-value. The UCB-bonus is defined as $r_t^{\mathrm{ucb}} = \beta \cdot \big[\eta(s_t, a_t)^\top \Lambda_t^{-1} \eta(s_t, a_t)\big]^{1/2}$, where $\beta$ is a constant, $\Lambda_t = \sum_{i=0}^m \eta(x_t^i, a_t^i)\eta(x_t^i, a_t^i)^\top + \lambda \cdot \mathbf{I}$ is the Gram matrix, and $m$ is the index of the current episode. The UCB-bonus measures the epistemic uncertainty of the state-action and is provably efficient [24].

For linear MDPs, we consider representation $z \in \mathbb{R}^c$ as the mean of the posterior $g^Z$ from the DB model, and set $z_t$ to be a linear function of the state-action encoding, i.e., $z_t = W_t \eta(s_t, a_t)$ parameterized by $W_t \in \mathbb{R}^{c \times d}$. Then, the following theorem establishes a connection between the DB-bonus $r^{\mathrm{db}}$ and the UCB-bonus $r^{\mathrm{ucb}}$.

**Theorem 1.** *In linear MDPs, for tuning parameter $\beta_0 > 0$, it holds that*

$$
\beta_0/\sqrt{2} \cdot r_t^{\mathrm{ucb}} \leq I(W_t; (s_t, a_t, S_{t+1})|\mathcal{D}_m)^{1/2} \leq \beta_0 \cdot r_t^{\mathrm{ucb}},
\tag{10}
$$

*where $I(W_t; (s_t, a_t, S_{t+1})|\mathcal{D}_m)^{1/2}$ is the DB-bonus $r^{\mathrm{db}}(s_t, a_t)$ under the linear MDP setting.*

In addition, using $r^{\mathrm{db}}$ as bonus leads to the same regret as LSVI-UCB by following a similar proof to Jin et al. [24]. We refer to Appendix C.2 for the problem setup and the detailed proofs. We remark that Theorem 1 is an approximate derivation because we only consider the predictive objective $I_{\mathrm{pred}}$ in (8) in Theorem 1. Nevertheless, introducing the contrastive objective $I_{\mathrm{nce}}$ is important in the training of the DB model as it prevents the mode collapse issue. Theorem 1 shows that the DB-bonus provides an instantiation of the UCB-bonus in DRL, which enables us to measure the epistemic uncertainty of high-dimensional states and actions without the linear MDP assumption.

**Connection to visiting count in tabular MDP**   The following theorem establishes connections between DB-bonus and the count-based bonus $r^{\mathrm{count}}(s_t, a_t) = \frac{\beta}{\sqrt{N_{s_t,a_t}+\lambda}}$ in tabular MDPs.

**Theorem 2.** *In tabular MDPs, it holds for the DB-bonus $r^{\mathrm{db}}(s_t, a_t)$ and the count-based intrinsic reward $r^{\mathrm{count}}(s_t, a_t)$ that,*

$$r^{\mathrm{db}}(s_t, a_t) \approx \frac{\sqrt{|\mathcal{S}|/2}}{\sqrt{N_{s_t,a_t}+\lambda}} \;=\; \beta_0 \cdot r^{\mathrm{count}}(s_t, a_t), \qquad (11)$$

*when $N_{s_t,a_t}$ is large, where $\lambda > 0$ is a tuning parameter, $|\mathcal{S}|$ is the number of states in tabular setting.*

We refer to Appendix C.3 for a detailed proofs. As a result, DB-bonus can also be considered as a count-based intrinsic reward in the space of dynamics-relevant representations.

## 4.2   Empirical Estimation

To estimate such a bonus under our DB model, we face several challenges. (i) Firstly, estimating the bonus defined in (9) requires us to parameterize representation under a Bayesian learning framework, whereas our DB model is parameterized by non-Bayesian neural networks. (ii) Secondly, estimating the DB-bonus defined in (9) requires us to compute the mutual information between the unknown transitions and the estimated model, which is in general hard as we do not have access to such transitions in general. To address such challenges, we estimate a lower bound of the DB-bonus, which is easily implementable and achieves reasonable performance empirically. Specifically, we consider to use $r_l^{\mathrm{db}}(s_t, a_t)$ as the lower bound of the information gain in (9),

$$r^{\mathrm{db}}(s_t, a_t) \geq \Big[ \mathcal{H}\big(g(s_t, a_t, S_{t+1})|\mathcal{D}_m\big) - \mathcal{H}\big(g(s_t, a_t, S_{t+1})|\Theta, \mathcal{D}_m\big) \Big]^{1/2} \triangleq r_l^{\mathrm{db}}(s_t, a_t), \qquad (12)$$

which holds for any mapping $g$ according to Data Processing Inequality (DPI). DPI is an information theoretic concept that can be understood as 'post-processing' cannot increase information. Since $g(s_t, a_t, S_{t+1})$ is a post-processing of $(s_t, a_t, S_{t+1})$, we have $I(\Theta; (s_t, a_t, S_{t+1})) > I(\Theta; g(s_t, a_t, S_{t+1}))$, where $g$ is a neural network in practice. In our model, we adopt the following mapping,

$$g(s_t, a_t, S_{t+1})|\Theta, \mathcal{D}_m = g^Z(s_t, a_t; \phi), \qquad (13)$$

where $g^Z$ is the representation distribution of DB, and $\phi$ constitutes a part of parameters of the total parameters $\Theta$. Intuitively, since $g^Z$ is trained by IB principle to capture information of transitions, adopting the mapping $g^Z$ to (12) yields a reasonable approximation of the DB-bonus. It further holds

$$r_l^{\mathrm{db}}(s_t, a_t) = \Big[ \mathcal{H}\big(g^{\mathrm{margin}}\big) - \mathcal{H}\big(g^Z(s_t, a_t; \phi)\big) \Big]^{1/2} = \mathbb{E}_\Theta D_{\mathrm{KL}}\big[ g^Z(z_t|s_t, a_t; \phi)\|g^{\mathrm{margin}}\big]^{1/2}, \quad (14)$$

where we define $g^{\mathrm{margin}} = g(s_t, a_t, S_{t+1})|\mathcal{D}_m$ as the marginal of the encodings over the posterior of the parameters $\Theta$ of the DB model. In practice, since $g^{\mathrm{margin}}$ is intractable, we approximate $g^{\mathrm{margin}}$ with standard Gaussian distribution. We remark that such approximation is motivated by the training of DB model, which drives the marginal of representation $g^Z$ toward $\mathcal{N}(0, \mathbf{I})$ through minimizing $I_{\mathrm{upper}}$ in (7). Such approximation leads to a tractable estimation and stable empirical performances.

In addition, since we do not train the DB model with Bayesian approach, we replace the expectation over posterior $\Theta$ in (14) by the corresponding point estimation, namely the parameter $\Theta$ of the neural networks trained with DB model on the dataset $\mathcal{D}_m$. To summarize, we utilize the following approximation of the DB-bonus $r^{\mathrm{db}}$ proposed in (9),

$$\hat{r}_l^{\mathrm{db}}(s_t, a_t) = D_{\mathrm{KL}}\big[ g^Z(\cdot|s_t, a_t; \phi) \,\|\, \mathcal{N}(0, \mathbf{I})\big]^{1/2} \approx r_l^{\mathrm{db}}(s_t, a_t). \qquad (15)$$

Since DB is trained by IB principle, which filters out the dynamics-irrelevant information, utilizing the bonus defined in (15) allows the agent to conduct robust exploration in noisy environments.

We summarize the the overall RL algorithm with self-supervised exploration induced by the DB-bonus in Algorithm 1, which we refer to as Self-Supervised Exploration with DB-bonus (SSE-DB). For the RL implementation, we adopt Proximal Policy Optimization (PPO) [45] with generalized advantage estimation [44] and the normalization schemes from Burda et al. [11]. We refer to Appendix D for the implementation details. The codes are available at `https://github.com/Baichenjia/DB`.

**Algorithm 1** SSE-DB

1: **Initialize:** The DB model and the actor-critic network
2: **for** episode $i = 1$ to $M$ **do**
3:     **for** timestep $i = 0$ to $T - 1$ **do**
4:         Obtain action from the actor $a_t = \pi(s_t)$, then execute $a_t$ and observe the state $s_{t+1}$;
5:         Add $(s_t, a_t, s_{t+1})$ into the on-policy experiences;
6:         Obtain the DB-bonus $\hat{r}_l^{\mathrm{db}}$ of $(s_t, a_t)$ by (15);
7:     **end for**
8:     Update the actor and critic by PPO with the collected on-policy experiences as the input;
9:     Update DB by gradient descent based on (8) with the collected on-policy experiences;
10: **end for**

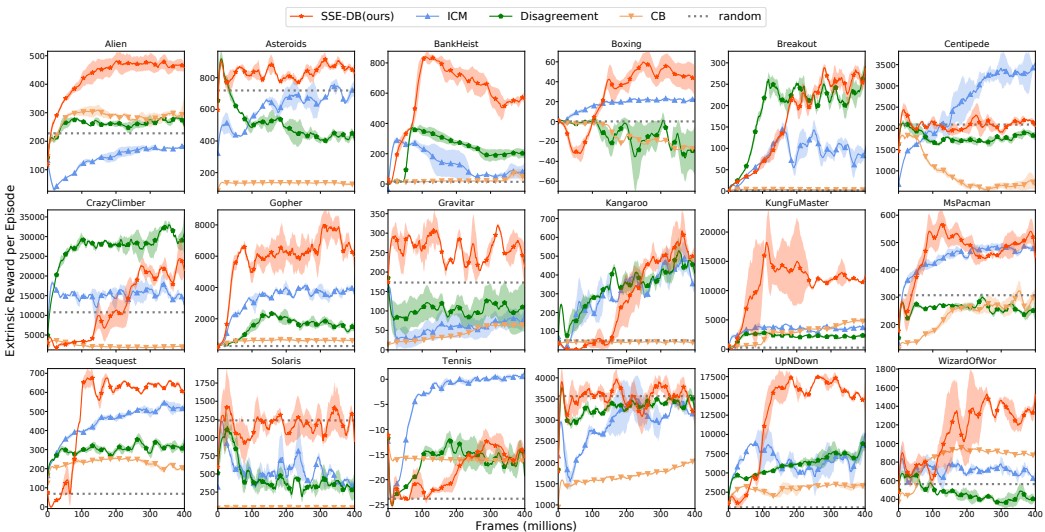

Figure 3: The evaluation curve in Atari games. The different methods are trained with different intrinsic rewards. The extrinsic rewards are only used to measure the performance. Each method was run with three random seeds.

## 5 Experiments

We evaluate SSE-DB on Atari games. We conduct experiments to compare the following methods. (i) **SSE-DB**. The proposed method in Alg. 1. (ii) **Intrinsic Curiosity Model (ICM)** [39]. ICM uses an inverse dynamics model to extract features related to the actions. ICM further adopts the prediction error of dynamics as the intrinsic reward for exploration. (iii) **Disagreement** [40]. This method captures epistemic uncertainty by the disagreement among predictions from an ensemble of dynamics models. Disagreement performs competitive to ICM and RND [12]. Also, this method is robust to white-noise. (iv) **Curiosity Bottleneck (CB)** [26]. CB quantifies the compressiveness of observation with respect to the representation as the bonus. CB is originally proposed for exploration with extrinsic rewards. We adapt CB for self-supervised exploration by setting the extrinsic reward zero. We compare the model complexity of all the methods in Appendix D. Other methods including Novelty Search [53] and Contingency-aware exploration [16] are also deserve to compare. However, we find Novelty Search ineffective in our implementation since the detailed hyper-parameters and empirical results in Atari are not available. Contingency-aware exploration is related to DB while the attention module is relatively complicated and the code is not achievable.

### 5.1 The Main Results

We evaluate all methods on Atari games with high-dimensional observations. The selected 18 games are frequently used in previous approaches for efficient exploration. The overall results are provided in Fig. 3. We highlight that in our experiments, the agents are trained without accessing the extrinsic

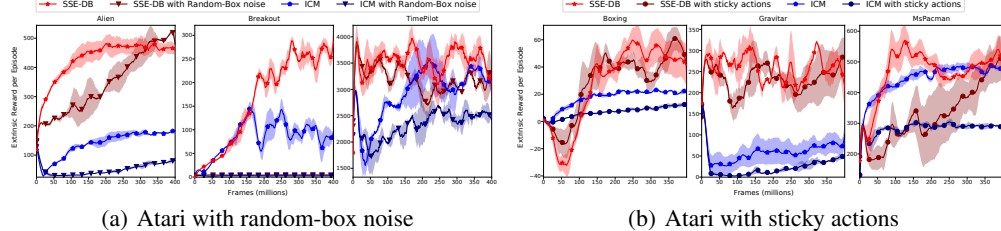

| | (a) Atari with random-box noise | | | (b) Atari with sticky actions |

(a) Atari with random-box noise        (b) Atari with sticky actions

Figure 5: A comparison on selected Atari games with and without noises. (a) Random-box noise. In Alien and TimePilot, SSE-DB is barely affected by random-box noise. Both methods fail in Breakout. (b) Sticky actions. Illustration shows SSE-DB is barely affected by the sticky actions.

rewards. The extrinsic rewards are ONLY utilized to evaluate the performance of the policies obtained from self-supervised exploration. Our experiments show that SSE-DB performs the best in 15 of 18 tasks, suggesting that dynamics-relevant feature together with DB-bonus helps the exploration of states with high extrinsic rewards.

In addition, since pure exploration without extrinsic rewards is very difficult in most tasks, a random baseline is required to show whether the exploration methods learn meaningful behaviors. We adopt the random score from DQN [35] and show the comparison in the figure. In Solaris, Centipede and TimePilot, our method obtains similar scores to random policy, which suggests that relying solely on intrinsic rewards is insufficient to solve these tasks. We also observe that SSE-DB is suboptimal in Tennis. A possible explanation is that for Tennis, the prediction error based methods, such as ICM, could capture additional information in the intrinsic rewards. For example, in Tennis, the prediction error becomes higher when the ball moves faster or when the agent hits the ball towards a tricky direction. The prediction-error based methods can naturally benefit from such nature of the game. In contrast, SSE-DB encourages exploration based on the information gain from learning the dynamics-relevant representation, which may not capture such critical events in Tennis.

## 5.2 Robustness in the Presence of Noises

**Observation Noises.** To analyze the robustness of SSE-DB to observation noises, an important evaluation metric is the performance of SSE-DB in the presence of dynamics-irrelevant information. A particularly challenging distractor is the white-noise [11, 26], which incorporates random task-irrelevant patterns to the observations. In such a scenario, a frequently visited state by injecting an unseen noise pattern may be mistakenly assigned with a high intrinsic reward by curiosity or pseudo-count based methods.

We use two types of distractors for the observations of Atari games, namely, (1) the random-box noise distractor, which places boxes filled with random Gaussian noise over the raw pixels, and (2) the pixel-level noise distractor, which adds pixel-wise Gaussian noise to the observations. Fig. 4 shows examples of the two types of distractors. In the sequel, we discuss results for the random-box noise distractor on selected Atari games, which we find sufficiently representative, and defer the complete report to Appendix E.1 and E.2.

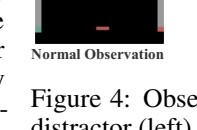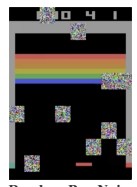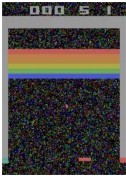

**Normal Observation**    **Random-Box Noise**    **Pixel Noise**

Figure 4: Observation of Breakout with no distractor (left), random-box noise distractor (middle), and pixel noise distractor (right).

Fig. 5(a) shows the performance of the compared methods on Alien, Breakout and TimePilot with and without noises. We observe that SSE-DB outperforms ICM on Alien and TimePilot with random-box noises. Nevertheless, in Breakout, we observe that both the methods fail to learn informative policies. A possible explanation is that, in Breakout, the ball is easily masked by the box-shaped noise (i.e., middle of Fig. 4). The random-box noise therefore buries critical transition information of the ball, which hinders all the baselines to extract dynamics-relevant information and leads to failures on Breakout with random-box noise as shown in Fig. 5(a).

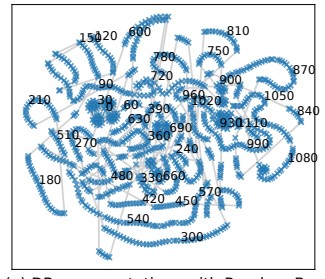
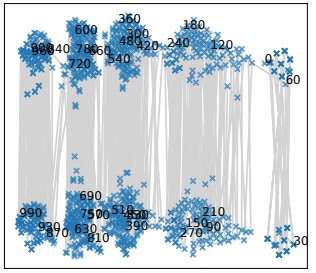

(a) DB representations with Random-Box      (b) ICM representations with Random-Box

Figure 6: Visualization of latent representations of MsPacman with random-box noise. The figures plot the representation of the same trajectory by DB (left) and ICM (right). Consecutive states are connected by the shaded lines. Numbers on the representations are the corresponding number of time steps. Representation learned by DB tends to align consecutive states on the same curve.

**Action Noise.** In addition to observation noises, noises in actions also raise challenges for learning the transition dynamics. To further study the robustness of SSE-DB, we conduct experiments on Atari games with sticky actions [32, 40]. At each time step, the agent may execute the previous action instead of the output action of current policy with a probability of $0.25$. We illustrate results on three selected Atari games, i.e., Boxing, Gravitar and MsPacman, in Fig. 5(b) and defer the complete results to Appendix E.3. Our experiments show that SSE-DB is robust to action noises, whereas ICM suffers significant performance drop from the action noise.

### 5.3 Visualization and Ablation Study

**Visualization of the Learned Representations.** To understand the latent representation learned by the DB model, we visualize the learned $Z$ with t-SNE [31] plots, which projects a 128d $z$-vector to a 2d one through dimensionality reduction. We compare the representations learned by SSE-DB and ICM with random-box noise. We illustrate the learned representations of MsPacman in Fig. 6. According to the visualization, representations learned by DB tends to align temporally-consecutive movements on the same curve. Moreover, each segment of a curve corresponds to a semantic component of the trajectory, such as eating pellets aligned together or avoiding ghosts. The segments of a curve end up with critical states, including death and reborn of the agent and ghosts. The visualization indicates that DB well captures the dynamics-relevant information in the learned representations. In contrast, such temporally-consecutive patterns are missing in the learned representation of ICM.

**Visualization of the DB bonus.** We provide visualization of the DB-bonus in Appendix E.5. The results show the DB-bonus effectively encourages the agent to explore the informative transitions.

**Ablation Study.** The training of DB loss consists of multiple components, including $I_{\mathrm{pred}}$, $I_{\mathrm{nce}}$, $I_{\mathrm{upper}}$, and the momentum observation encoder. To analyze the importance of the components, we conduct an ablation study by removing each of them respectively and evaluate the DB model correspondingly. The ablation study suggests that the all the components are crucial for learning effective dynamics-relevant representations. In addition, we observe that $I_{\mathrm{upper}}$ is particularly important in the environment with dynamics-irrelevant noise. Please refer to Appendix E.4 for details.

## 6 Conclusion

In this paper, we introduce Dynamic Bottleneck model that learns dynamics-relevant representations based on the IB principle. Based on the DB model, we further propose DB-bonus based on the DB model for efficient exploration. We establish theoretical connections between the proposed DB-bonus and provably efficient bonuses. Our experiments show that SSE-DB outperforms several strong baselines in stochastic environments for self-supervised exploration. Moreover, we observe that DB learns well-structured representations and the DB-bonus characterizes informative transitions for exploration. For our future work, we wish to combine the DB representation to effective exploration methods including BeBold [61] and NGU [6] to enhance their robustness in stochastic environments.

## Acknowledgements

The authors thank Tencent Robotics X and Vector Institute for the computation resources supported. Part of the work was done during internship at Tencent Robotics X lab. The authors also thank the anonymous reviewers, whose invaluable suggestions have helped us to improve the paper.

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
