# A  Proof of Equation 5

*Proof.* We introduce the latent variable $C$ to indicate whether the next-state encoding $s_{t+1}$ the representation $z_t$ are drawn from the joint density ($C = 1$) or from the product of marginals ($C = 0$). For the positive sample with $C = 1$, we have

$$p(z_t, s_{t+1}|C = 1) = p(z_t, s_{t+1}),$$

which is the joint density. For the negative example, we have

$$p(z_t, s_{t+1}|C = 0) = p(z_t)p(s_{t+1}),$$

which is the product of marginals. In the sequel, we use contrastive objective containing one positive pair and $N$ negative pairs. Correspondingly, the priors of latent $C$ takes the form of

$$p(C = 1) = 1/(N + 1), \quad p(C = 0) = N/(N + 1).$$

The following computation is adopted from [36]. By Bayesian rule, the posterior of $C = 1$ takes the form of

$$
\begin{aligned}
&\log p(C = 1|z_t, s_{t+1}) \\
&= \log \frac{p(C = 1)p(z_t, s_{t+1}|C = 1)}{p(C = 0)p(z_t, s_{t+1}|C = 0) + p(C = 1)p(z_t, s_{t+1}|C = 1)} \\
&= \log \frac{p(C = 1)p(z_t, s_{t+1})}{p(C = 0)p(z_t)p(s_{t+1}) + p(C = 1)p(z_t, s_{t+1})} \\
&= \log \frac{p(z_t, s_{t+1})}{Np(z_t)p(s_{t+1}) + p(z_t, s_{t+1})} \\
&= -\log(1 + N\frac{p(z_t)p(s_{t+1})}{p(z_t, s_{t+1})}) \\
&\le -\log N + \log \frac{p(z_t, s_{t+1})}{p(z_t)p(s_{t+1})}.
\end{aligned}
\tag{16}
$$

By taking expectation with respect to $p(z_t, s_{t+1}|C = 1)$ on both sides of (16) and rearranging, we obtain

$$I(Z_t, S_{t+1}) \ge \log N + \mathbb{E}_{p(z_t, s_{t+1})}[\log p(C = 1|z_t, s_{t+1})], \tag{17}$$

where $I(Z_t; S_{t+1}) = \mathbb{E}_{p(z_t, s_{t+1})}[\log \frac{p(z_t, s_{t+1})}{p(z_t)p(s_{t+1})}]$ is the mutual information between the distributions of the dynamics-relevant representation $Z_t$ and next-observation encoding $S_{t+1}$. In the sequel, we follow [36] and fit a score function $h(z_t, s_{t+1})$ to maximize the log-likelihood of positive samples by solving a binary classification problem,

$$\mathcal{L}_{\mathrm{nce}}(h) = \mathbb{E}_{p(z_t, s_{t+1})}\mathbb{E}_{S^-}\left[\log \frac{\exp(h(z_t, s_{t+1}))}{\sum_{s_j \in S^- \cup s_{t+1}} \exp(h(z_t, s_j))}\right]. \tag{18}$$

where the denominator involves both the positive and negative pairs. If $h$ is sufficiently expressive, the optimal solution of the binary classifier is $h^*(z_t, s_{t+1}) = p(C = 1|z_t, s_{t+1})$. Thus, we have

$$
\begin{aligned}
I(Z_t, S_{t+1}) &\ge \log N + \mathbb{E}_{p(z_t, s_{t+1})}[\log p(C = 1|z_t, s_{t+1})] \\
&= \log N + \mathbb{E}_{p(z_t, s_{t+1})}[\log h^*(z_t, s_{t+1})] \\
&\ge \log N + \mathbb{E}_{p(z_t, s_{t+1})}[\log h^*(z_t, s_{t+1}) - \log \sum_{s_j \in S^- \cup s_{t+1}} \exp(h^*(z_t, s_j))] \\
&= \log N + \mathcal{L}_{\mathrm{nce}}(h^*) = \log N + \max_h \mathcal{L}_{\mathrm{nce}}(h) \\
&\ge \log N + \mathcal{L}_{\mathrm{nce}}(h).
\end{aligned}
\tag{19}
$$

The third line holds since $h^*(z_t, s_{t+1}) \in [0, 1]$ and, hence, the added term is strictly negative. Since $N$ is a constant decided by the training batch in DB, it suffices to maximize $\mathcal{L}_{\mathrm{nce}}(h)$ in (19) to maximize the mutual information $I(Z_t, S_{t+1})$. $\quad\square$

# B   Pseudocode for DB training

The network of DB in Fig. 2 contains four modules, we introduce the network architecture as follows.

- *Observation Encoder $f_o^S(\cdot; \theta_o)$ and $f_m^S(\cdot; \theta_m)$.* The observation encoder contains a main network and a momentum network, which are used to extract features of $o_t$ and $o_{t+1}$, respectively. Each network contains three convolution layers as Conv(filter=32, kernel-size=8, strides=4) $\to$ Conv(filter=64, kernel-size=4, strides=2)$\to$ Conv(filter=64, kernel-size=3, strides=1). We adopt Leaky ReLU as teh activation function. The architecture is similar to DQN without FCNs [35]. The input to observation encoder is image and the output is a vector $s \in \mathbb{R}^{512}$.

- *Representation Posterior $g^Z(\cdot; \phi)$.* The posterior $g^Z(s_t, a_t; \phi)$ represents the dynamics-relevant information extracted from state and action. We concatenate $s_t \in \mathbb{R}^{512}$ and one-hot vector $a_t \in \mathbb{R}^{|\mathcal{A}|}$ as the input. The processing flow is FCN(units=256) $\to$ ResNet(units=(256,256)) $\to$ ResNet(units=(256,256)) $\to$ FCN(units=256), where ResNet is a residual network with two layers. The output of $g^Z$ is a diagonal Gaussian with mean $\mu_z \in \mathbb{R}^{128}$ and variance $\sigma_z \in \mathbb{R}^{128}$. We use soft-plus activation to make the variance positive. The representation $z \in \mathbb{R}^{128} \sim \mathcal{N}(\mu_z, \sigma_z)$.

- *Prediction Head $q(\cdot; \psi)$.* The prediction head $q(z; \psi)$ contains three ResNets and two FCNs. We process the input $z \in \mathbb{R}^{128}$ by FCN(units=256) $\to$ ResNet(units=(256,256)) $\to$ ResNet(units=(256,256)) $\to$ FCN(units=512) $\to$ ResNet(units=(512,512)), where ResNet is a residual network with two layers. The output is a mean vector, which has the same dimensions as $s_{t+1}$, and an additional variance estimation. The output of $q_\psi$ is a diagonal Gaussian that has the same variance in each dimension.

- *Projection head $f_o^P(\cdot; \varphi_o)$ and $f_m^P(\cdot; \varphi_m)$.* The projection heads map $\bar{q}(z_t)$ and $s_{t+1}$ to low-dimensional space for contrastive estimation. The projection head contains two FCNs with 256 and 128 units, respectively. We follow Chen et al. [14] and adopt a normalization layer at each layer.

---

**Algorithm 2** Pseudocode for DB training, PyTorch-like

---

```
# fₒˢ, fₘˢ: observation encoders, gᶻ: representation posterior, q: prediction head
# fₒᴾ, fₘᴾ: projection heads, 𝒲: weight matrix in contrastive loss

for (o, a, o_next) in loader: # Load 16 episodes from 16 actors. N=128*16-1.
    s, s_next = fₒˢ(o), fₘˢ(o_next).detach() # observation encoder
    z_dis = gᶻ(s,a) # Gaussian distribution of Z
    I_upper = KL(z_dis, N(0,I)) # KL-divergence to compress representation (a upper
        bound)

    z = r_sample(z_dis) # reparameterization
    s_pred_dis = q(z) # prediction head, the output is a Gaussian distribution
    I_pred = log(s| s_pred_dis) # predictive objective (lower bound)

    s_pred = mean(s_pred_dis) # take the mean value of prediction
    s_pred_proj = fₒᴾ(s_pred) # projection head
    s_next_proj = fₘᴾ(s_next).detach() # momentum projection head

    logits = matmul(s_pred_proj, matmul(𝒲, s_next_proj.T)) # (N-1) x (N-1)
    logits = logits - max(logits, axis=1) # subtract max from logits for stability
    labels = arange(logits.shape[0])
    I_nce = -CrossEntropyLoss(logits, labels) # contrastive estimation (lower bound)

    L = α₁·I_upper - α₂·I_pred - α₃·I_nce # total loss function
    L.backward() # back-propagate
    update(fₒˢ, gᶻ, q, fₒᴾ, 𝒲) # Adam update the parameters
    θ_m = τ· θ_m+(1-τ) ·θ_o
    θ_m = τ· φ_m+(1-τ) ·φ_m
```

---

We use 128 actors in experiments, and each episode contains 128 steps. Since the batch size 128*128 is too large for GPU memory (RTX-2080Ti), we follow the implementation of ICM and Disagreement by using experiences from 16 actors for each training step. We iterate 8 times to sample all experiences from 128 actors. As a result, the corresponding negative sample size $|S^-|$ is $16 * 128 - 1$ for contrastive estimation.

## C  Proof of the DB-bonus

### C.1  Background: LSVI-UCB

The algorithmic description of LSVI-UCB [24] is given in Algorithm 3. The feature map of the state-action pair is denoted as $\eta : \mathcal{S} \times \mathcal{A} \to \mathbb{R}^d$. The transition kernel and reward function are assumed to be linear in $\eta$. Under such a setting, it is known that for any policy $\pi$, the corresponding action-value function $Q_t(s,a) = \chi_t^\top \eta(s,a)$ is linear in $\eta$ [24]. Each iteration of LSVI-UCB consists of two parts. First, in line 3-6, the agent executes the policy according to $Q_t$ for an episode. Second, in line 7-11, the parameter $\chi_t$ of $Q$-function is updated in closed-form by following the regularized least-squares

$$\chi_t \leftarrow \arg\min_{\chi \in \mathbb{R}^d} \sum_{i=0}^{m} \big[ r_t(s_t^i, a_t^i) + \max_{a \in \mathcal{A}} Q_{t+1}(s_{t+1}^i, a) - \chi^\top \eta(s_t^i, a_t^i) \big]^2 + \lambda \|\chi\|^2,$$

where $m$ is the number of episodes, and $i$ is the index of episodes. The least-squares problem has the following closed form solution ,

$$\chi_t = \Lambda_t^{-1} \sum_{\tau=0}^{m} \eta(x_t^i, a_t^i) \big[ r_t(x_t^i, a_t^i) + \max_a Q_{t+1}(x_{t+1}^i, a) \big],$$

where $\Lambda_t$ is the Gram matrix. The action-value function is estimated by $Q_t(s,a) \approx \chi_t^\top \eta(s,a)$.

LSVI-UCB uses UCB-bonus (line 10) to construct the confidence bound of $Q$-function as $r^{\mathrm{ucb}} = \beta \big[ \eta(s,a)^\top \Lambda_t^{-1} \eta(s,a) \big]^{1/2}$ [1], which measures the epistemic uncertainty of the corresponding state-action pairs. Theoretical analysis shows that LSVI-UCB achieves a near-optimal worst-case regret of $\tilde{\mathcal{O}}(\sqrt{d^3 T^3 L^3})$ with proper selections of $\beta$ and $\lambda$, where $L$ is the total number of steps. We refer to Jin et al. [24] for the detailed analysis.

---

**Algorithm 3** LSVI-UCB for linear MDP

---

1: **Initialize:** $\Lambda_t \leftarrow \lambda \cdot \mathbf{I}$ and $w_h \leftarrow 0$
2: **for** episode $m = 0$ **to** $M - 1$ **do**
3:     Receive the initial state $s_0$
4:     **for** step $t = 0$ **to** $T - 1$ **do**
5:         Take action $a_t = \arg\max_{a \in \mathcal{A}} Q_t(s_t, a)$ and observe $s_{t+1}$.
6:     **end for**
7:     **for** step $t = T - 1$ **to** $0$ **do**
8:         $\Lambda_t \leftarrow \sum_{i=0}^{m} \eta(x_t^i, a_t^i) \eta(x_t^i, a_t^i)^\top + \lambda \cdot \mathbf{I}$
9:         $\chi_t \leftarrow \Lambda_t^{-1} \sum_{i=0}^{m} \eta(x_t^i, a_t^i) [r_t(x_t^i, a_t^i) + \max_a Q_{t+1}(x_{t+1}^i, a)]$
10:         $Q_t(\cdot, \cdot) = \min\{\chi_t^\top \eta(\cdot, \cdot) + \alpha[\eta(\cdot, \cdot)^\top \Lambda_t^{-1} \eta(\cdot, \cdot)]^{1/2}, T\}$
11:     **end for**
12: **end for**

---

### C.2  Proof of connection to LSVI-UCB

In linear function approximation, we set the representation of DB to be linear in state-action encoding, namely, $z_t = W_t \eta(s_t, a_t) \in \mathbb{R}^c$, where $W_t \in \mathbb{R}^{c \times d}$ and $\eta(s_t, a_t) \in \mathbb{R}^d$. To capture the prediction error in DB, we conduct regression to recover the next state $s_{t+1}$ and consider the following regularized least-square problem,

$$w_t \leftarrow \arg\min_{W} \sum_{i=0}^{m} \big\| s_{t+1}^i - W_t \eta(s_t^i, a_t^i) \big\|_F^2 + \lambda \|W\|_F^2, \tag{20}$$

where $\| \cdot \|_F$ denotes the Frobenius norm. In the sequel, we consider a Bayesian linear regression perspective of (20) that captures the intuition behind the DB-bonus. Our objective is to approximate the next-state prediction $s_{t+1}$ via fitting the parameter $W$, such that

$$W\eta(s_t, a_t) \approx s_{t+1},$$

where $s_{t+1}$ is given. We assume that we are given a Gaussian prior of the initial parameter $W \sim \mathcal{N}(\mathbf{0}, \mathbf{I}/\lambda)$. With a slight abuse of notation, we denote by $W_t$ the Bayesian posterior of the parameter

$W$ given the set of independent observations $\mathcal{D}_m = \{(s_t^i, a_t^i, s_{t+1}^i)\}_{i \in [0,m]}$. We further define the following noise with respect to the least-square problem in (20),

$$\epsilon = s_{t+1} - W_t \eta(s_t, a_t) \in \mathbb{R}^c, \tag{21}$$

where $(s_t, a_t, s_{t+1})$ follows the distribution of trajectory. The following theorem justifies the DB-bonus under the Bayesian linear regression perspective.

**Theorem 3** (Formal Version of Theorem 1). *We assume that $\epsilon$ follows the standard multivariate Gaussian distribution $\mathcal{N}(0, \mathbf{I})$ given the state-action pair $(s_t, a_t)$ and the parameter $W$. Assuming $W$ follows the Gaussian prior $\mathcal{N}(0, \mathbf{I}/\lambda)$. We define*

$$\Lambda_t = \sum_{i=0}^{m} \eta(x_t^i, a_t^i) \eta(x_t^i, a_t^i)^\top + \lambda \cdot \mathbf{I}. \tag{22}$$

*It then holds for the posterior of $W_t$ given the set of independent observations $\mathcal{D}_m = \{(s_t^i, a_t^i, s_{t+1}^i)\}_{i \in [0,m]}$ that*

$$\sqrt{\frac{c}{4}} \left[ \eta(t)^\top \Lambda_t^{-1} \eta(t) \right]^{1/2} \leq I(W_t; [s_t, a_t, S_{t+1}] | \mathcal{D}_m)^{1/2} \leq \sqrt{\frac{c}{2}} \left[ \eta(t)^\top \Lambda_t^{-1} \eta(t) \right]^{1/2}.$$

*Proof.* The proof follows the standard analysis of Bayesian linear regression. See, e.g., West [57] for the details. We introduce the following notations for $W_t \in \mathbb{R}^{c \times d}$, $\eta(s_t, a_t) \in \mathbb{R}^d$, and $W_t \eta(s_t, a_t) \in \mathbb{R}^c$,

$$W_t = \begin{bmatrix} w_{11} & \cdots & w_{1d} \\ \vdots & & \vdots \\ w_{c1} & \cdots & w_{cd} \end{bmatrix} \in \mathbb{R}^{c \times d}, \quad \eta(s_t, a_t) = \begin{bmatrix} \eta_1 \\ \eta_2 \\ \vdots \\ \eta_d \end{bmatrix} \in \mathbb{R}^d, \quad W_t \eta(s_t, a_t) = \begin{bmatrix} \sum_{k=1}^d w_{1k} \eta_k \\ \sum_{k=1}^d w_{2k} \eta_k \\ \vdots \\ \sum_{k=1}^d w_{ck} \eta_k \end{bmatrix} \in \mathbb{R}^c. \tag{23}$$

For the analysis, we vectorize matrix $W_t$ and define a new matrix $\tilde{\eta}$. Meanwhile, we define $\tilde{\eta}$ by repeating $\eta(s_t, a_t)$ for $c$ times in the diagonal. Specifically, we define $\text{vec}(W_t) \in \mathbb{R}^{cd}$ and $\tilde{\eta}(s_t, a_t) \in \mathbb{R}^{cd \times c}$ as follows,

$$\text{vec}(W_t) = \begin{bmatrix} w_{11} \\ \vdots \\ w_{1d} \\ w_{21} \\ \vdots \\ w_{2d} \\ \vdots \\ w_{c1} \\ \vdots \\ w_{cd} \end{bmatrix} \in \mathbb{R}^{cd}, \quad \tilde{\eta}(s_t, a_t) = \begin{bmatrix} \eta(s_t, a_t) & 0 & \cdots & 0 \\ 0 & \eta(s_t, a_t) & \cdots & 0 \\ \vdots & \vdots & \ddots & \vdots \\ 0 & 0 & \cdots & \eta(s_t, a_t) \end{bmatrix} = \begin{bmatrix} \eta_1 & 0 & \cdots & 0 \\ \vdots & & & \\ \eta_d & 0 & \cdots & 0 \\ \eta_1 & \cdots & 0 \\ \vdots & \vdots & & \vdots \\ \eta_d & \cdots & 0 \\ \vdots & \vdots & & \vdots \\ 0 & 0 & \cdots & \eta_1 \\ \vdots & \vdots & & \vdots \\ 0 & 0 & \cdots & \eta_d \end{bmatrix} \in \mathbb{R}^{cd \times c}, \tag{24}$$

Then we have $\text{vec}(W_t)^\top \tilde{\eta}(s_t, a_t) = W_t \eta(s_t, a_t)$ according to block matrix multiplication and (24). Formally,

$$\text{vec}(W_t)^\top \tilde{\eta}(s_t, a_t) = \begin{bmatrix} \sum_{k=1}^d w_{1k} \eta_k \\ \sum_{k=1}^d w_{2k} \eta_k \\ \vdots \\ \sum_{k=1}^d w_{ck} \eta_k \end{bmatrix}^\top = W_t \eta(s_t, a_t) \in \mathbb{R}^c. \tag{25}$$

Our objective is to compute the posterior density $W_t = W | \mathcal{D}_m$, where $\mathcal{D}_m = \{(s_t^i, a_t^i, s_{t+1}^i)\}_{i \in [0,m]}$ is the set of observations. The target of the linear regression is $s_{t+1}$. By the assumption that $\epsilon$ follows the standard Gaussian distribution, we obtain that

$$s_{t+1} | (s_t, a_t), W_t \sim \mathcal{N}\big(W_t \eta(s_t, a_t), \mathbf{I}\big). \tag{26}$$

Because we have $\mathrm{vec}(W_t)^\top \tilde{\eta}(s_t, a_t) = W_t \eta(s_t, a_t)$ according to (25), we have

$$s_{t+1}|(s_t, a_t), W_t \sim \mathcal{N}\big(\mathrm{vec}(W_t)^\top \tilde{\eta}(s_t, a_t), \mathbf{I}\big). \tag{27}$$

Recall that we have the prior distribution $W \sim \mathcal{N}(0, \mathbf{I}/\lambda)$, then the prior of $\mathrm{vec}(W) \sim \mathcal{N}(0, \mathbf{I}/\lambda)$. It holds from Bayes rule that

$$\log p(\mathrm{vec}(W_t)|\mathcal{D}_m) = \log p(\mathrm{vec}(W_t)) + \log p(\mathcal{D}_m|\mathrm{vec}(W_t)) + Const. \tag{28}$$

Plugging (27) and the probability density function of Gaussian distribution into (28) yields

$$\log p(\mathrm{vec}(W_t)|\mathcal{D}_m) = -\|\mathrm{vec}(W_t)\|^2/2 - \sum_{i=1}^{m} \|\mathrm{vec}(W_t)\tilde{\eta}(s_t^i, a_t^i) - s_{t+1}^i\|^2/2 + Const$$
$$= -(\mathrm{vec}(W_t) - \tilde{\mu}_t)^\top \tilde{\Lambda}_t^{-1}(\mathrm{vec}(W_t) - \tilde{\mu}_t)/2 + Const, \tag{29}$$

where we define

$$\tilde{\mu}_t = \tilde{\Lambda}_t^{-1} \sum_{i=0}^{m} \tilde{\eta}(s_t^i, a_t^i)s_{t+1}^i \in \mathbb{R}^{cd}, \qquad \tilde{\Lambda}_t = \sum_{i=0}^{m} \tilde{\eta}(s_t^i, a_t^i)\tilde{\eta}(x_t^i, a_t^i)^\top + \lambda \cdot \mathbf{I} \in \mathbb{R}^{cd \times cd}.$$

Thus, by (29), we obtain that $\mathrm{vec}(W_t) = W|\mathcal{D}_m \sim \mathcal{N}(\tilde{\mu}_t, \tilde{\Lambda}_t^{-1})$. We have the covariance matrix of $\mathrm{vec}(W_t)$ is

$$\mathrm{Var}(\mathrm{vec}(W_t)) = \tilde{\Lambda}_t^{-1} \tag{30}$$

The $\tilde{\Lambda}_t$ term accumulates previous $\tilde{\eta}(s_t^i, a_t^i)\tilde{\eta}(s_t^i, a_t^i)^\top$. Since we have

$$\tilde{\eta}(s_t^i, a_t^i)\tilde{\eta}(s_t^i, a_t^i)^\top = \begin{bmatrix} \eta(t) & 0 & \cdots & 0 \\ 0 & \eta(t) & \cdots & 0 \\ \vdots & \vdots & \ddots & \vdots \\ 0 & 0 & \cdots & \eta(t) \end{bmatrix} \begin{bmatrix} \eta(t)^\top & 0 & \cdots & 0 \\ 0 & \eta(t)^\top & \cdots & 0 \\ \vdots & \vdots & \ddots & \vdots \\ 0 & 0 & \cdots & \eta(t)^\top \end{bmatrix}$$
$$= \begin{bmatrix} \eta(t)\eta(t)^\top & 0 & \cdots & 0 \\ & \eta(t)\eta(t)^\top & \cdots & 0 \\ \vdots & \vdots & \ddots & \vdots \\ 0 & 0 & \cdots & \eta(t)\eta(t)^\top \end{bmatrix} \in \mathbb{R}^{cd \times cd}, \tag{31}$$

where $\eta(t) = \eta(s_t^i, a_t^i) \in \mathbb{R}^d$, and $\eta(t)\eta(t)^\top \in \mathbb{R}^{d \times d}$ repeats for $c$ times, we further expand the matrix $\tilde{\Lambda}_t$ as,

$$\tilde{\Lambda}_t = \sum_{i=0}^{m} \tilde{\eta}(s_t^i, a_t^i)\tilde{\eta}(x_t^i, a_t^i)^\top + \lambda\mathbf{I} = \begin{bmatrix} \sum \eta(t)\eta(t)^\top + \lambda I & 0 & \cdots & 0 \\ 0 & \sum \eta(t)\eta(t)^\top + \lambda I & \cdots & 0 \\ \vdots & \vdots & \ddots & \vdots \\ 0 & 0 & \cdots & \sum \eta(t)\eta(t)^\top + \lambda I \end{bmatrix}$$
$$= \begin{bmatrix} \Lambda_t & 0 & \cdots & 0 \\ 0 & \Lambda_t & \cdots & 0 \\ \vdots & \vdots & \ddots & \vdots \\ 0 & 0 & \cdots & \Lambda_t \end{bmatrix}, \tag{32}$$

where we follow the definition of $\Lambda_t$ in (22). Further, the mutual information

$$I(\mathrm{vec}(W_t); [s_t, a_t, S_{t+1}]|\mathcal{D}_m) = \mathcal{H}(\mathrm{vec}(W_t)|\mathcal{D}_m) - \mathcal{H}(\mathrm{vec}(W_t)|(s_t, a_t, S_{t+1}) \cup \mathcal{D}_m)$$
$$= \frac{1}{2}\log\det\big(\mathrm{Var}(\mathrm{vec}(W_t)|\mathcal{D}_m)\big) - \frac{1}{2}\log\det\big(\mathrm{Var}(\mathrm{vec}(W_t)|(s_t, a_t, S_{t+1}) \cup \mathcal{D}_m)\big). \tag{33}$$

Plugging (30) into (33), we have

$$I(\mathrm{vec}(W_t); [s_t, a_t, S_{t+1}]|\mathcal{D}_m) = \frac{1}{2}\log\det\big(\tilde{\Lambda}_t^{-1}\big) - \frac{1}{2}\log\det\big((\tilde{\Lambda}_t^\dagger)^{-1}\big)$$
$$= \frac{1}{2}\log\det\big(\tilde{\Lambda}_t^\dagger\big) - \frac{1}{2}\log\det\big(\tilde{\Lambda}_t\big)$$
$$= \frac{1}{2}\log\det\big(\tilde{\Lambda}_t + \tilde{\eta}(s_t, a_t)\tilde{\eta}(s_t, a_t)^\top\big) - \frac{1}{2}\log\det\big(\tilde{\Lambda}_t\big) \tag{34}$$
$$= \frac{1}{2}\log\det\big(\tilde{\eta}(s_t, a_t)^\top \tilde{\Lambda}_t^{-1}\tilde{\eta}(s_t, a_t) + \mathbf{I}\big),$$

where the last line follows Matrix Determinant Lemma. Then, plugging (32) into (34), we have

$$
\tilde{\eta}(s_t, a_t)^\top \tilde{\Lambda}_t^{-1} \tilde{\eta}(s_t, a_t) = 
\begin{bmatrix}
\eta(t)^\top & 0 & \cdots & 0 \\
0 & \eta(t)^\top & \cdots & 0 \\
\vdots & \vdots & \ddots & \vdots \\
0 & 0 & \cdots & \eta(t)^\top
\end{bmatrix}
\begin{bmatrix}
\Lambda_t^{-1} & 0 & \cdots & 0 \\
0 & \Lambda_t^{-1} & \cdots & 0 \\
\vdots & \vdots & \ddots & \vdots \\
0 & 0 & \cdots & \Lambda_t^{-1}
\end{bmatrix}
\begin{bmatrix}
\eta(t) & 0 & \cdots & 0 \\
0 & \eta(t) & \cdots & 0 \\
\vdots & \vdots & \ddots & \vdots \\
0 & 0 & \cdots & \eta(t)
\end{bmatrix}
$$

$$
= 
\begin{bmatrix}
\eta(t)^\top \Lambda_t^{-1} \eta(t) & 0 & \cdots & 0 \\
0 & \eta(t)^\top \Lambda_t^{-1} \eta(t) & \cdots & 0 \\
\vdots & \vdots & \ddots & \vdots \\
0 & 0 & \cdots & \eta(t)^\top \Lambda_t^{-1} \eta(t)
\end{bmatrix} \in \mathbb{R}^{c \times c}. \tag{35}
$$

Thus, we have

$$
I(\text{vec}(W_t); [s_t, a_t, S_{t+1}]|\mathcal{D}_m) = \frac{1}{2} \log \det \left( \tilde{\eta}(s_t, a_t)^\top \tilde{\Lambda}_t^{-1} \tilde{\eta}(s_t, a_t) + \mathbf{I} \right)
$$
$$
= \frac{c}{2} \cdot \log \left( \eta(t)^\top \Lambda_t^{-1} \eta(t) + 1 \right). \tag{36}
$$

By assuming the L2-norm of feature vector $\|\eta\|_2 \le 1$ and $\lambda = 1$ [24, 56], we have $\eta(t)^\top \Lambda_t^{-1} \eta(t) \le 1$. Moreover, since $\frac{x}{2} \le \log(1+x) \le x$ for $x \in [0, 1]$, we have

$$
\frac{\eta(t)^\top \Lambda_t^{-1} \eta(t)}{2} \le \log \left( \eta(t)^\top \Lambda_t^{-1} \eta(t) + 1 \right) \le \eta(t)^\top \Lambda_t^{-1} \eta(t). \tag{37}
$$

Therefore, we have

$$
\sqrt{\frac{c}{4}} \left[ \eta(t)^\top \Lambda_t^{-1} \eta(t) \right]^{1/2} \le I(\text{vec}(W_t); [s_t, a_t, S_{t+1}]|\mathcal{D}_m)^{1/2} \le \sqrt{\frac{c}{2}} \left[ \eta(t)^\top \Lambda_t^{-1} \eta(t) \right]^{1/2}. \tag{38}
$$

Moreover, since $\text{vec}(W_t)$ is the vectorization of $W_t$, we have $I(W_t; [S_t, A_t]|\mathcal{D}_m) = I(\text{vec}(W_t); [S_t, A_t]|\mathcal{D}_m)$. Finally, we have

$$
\sqrt{\frac{c}{4}} \left[ \eta(t)^\top \Lambda_t^{-1} \eta(t) \right]^{1/2} \le I(W_t; [s_t, a_t, S_{t+1}]|\mathcal{D}_m)^{1/2} \le \sqrt{\frac{c}{2}} \left[ \eta(t)^\top \Lambda_t^{-1} \eta(t) \right]^{1/2}. \tag{39}
$$

Recall that we set $r^{\text{ucb}} = \beta \left[ \eta(t)^\top \Lambda_t^{-1} \eta(t) \right]^{1/2}$. Thus, we obtain that

$$
\frac{1}{\sqrt{2}} \beta_0 \cdot r^{\text{ucb}} \le I(W_t; [s_t, a_t, S_{t+1}]|\mathcal{D}_m)^{1/2} \le \beta_0 \cdot r^{\text{ucb}}, \tag{40}
$$

where $\beta_0 = \sqrt{\frac{c}{2\beta^2}}$ is a tuning parameter. Thus, we complete the proof of Theorem 1. $\qquad\square$

### C.3  Proof of Theorem 2

*Proof.* In the sequel, we consider the tabular setting with finite state and action spaces. We define $d = |\mathcal{S}| \times |\mathcal{A}|$. In the tabular setting, we define the state-action encoding by the one-hot vector indexed by state-action pair $(s, a) \in \mathcal{S} \times \mathcal{A}$. For a state-action pair $(s_j, a_j) \in \mathcal{S} \times \mathcal{A}$, where $j \in [0, d-1]$, we have

$$
\eta(s_j, a_j) = 
\begin{bmatrix}
0 \\
\vdots \\
1 \\
\vdots \\
\vdots \\
0
\end{bmatrix}
\in \mathbb{R}^d, \qquad
\eta(s_j, a_j)\eta(s_j, a_j)^\top = 
\begin{bmatrix}
0 & \cdots & 0 & \cdots & 0 \\
\vdots & \ddots & & & \vdots \\
0 & & 1 & & 0 \\
\vdots & & & \ddots & \vdots \\
0 & \cdots & 0 & \cdots & 0
\end{bmatrix}
\in \mathbb{R}^{d \times d}, \tag{41}
$$

where the value is 1 at the $j$-th entry and 0 elsewhere. Moreover, the gram matrix $\Lambda_j = \sum_{i=0}^m \eta(x_j^i, a_j^i)\eta(x_j^i, a_j^i)^\top + \lambda \cdot \mathbf{I}$ is the sum of all the matrices $\eta(s_j, a_j)\eta(s_j, a_j)^\top$ corresponding to the batch $\mathcal{D}_m$. That said, we have

$$
\Lambda_j = 
\begin{bmatrix}
n_0 + \lambda & 0 & & \cdots & & 0 \\
0 & n_1 + \lambda & & \cdots & & 0 \\
\vdots & & \ddots & & & \vdots \\
0 & & & n_j + \lambda & & 0 \\
\vdots & & & & \ddots & \vdots \\
0 & & \cdots & & \cdots & n_{d-1} + \lambda
\end{bmatrix}, \tag{42}
$$

where the $j$-th diagonal element of $\Lambda_j$ is the corresponding counts for state-action $(s_j, a_j)$, i.e.,

$$n_j = N_{s_j, a_j}.$$

Thus, following the proof from §C.2, the mutual-information scales with $\log\left(\eta(t)^\top \Lambda_t^{-1} \eta(t) + 1\right)$. When the count $N_{s_j, a_j}$ of state-action pairs are large for all $j \in [0, d-1]$, we have $\log\left(\eta(t)^\top \Lambda_t^{-1} \eta(t) + 1\right) \approx \eta(t)^\top \Lambda_t^{-1} \eta(t)$. Moreover, it holds that

$$\left[\eta(s_j, a_j)^\top \Lambda_t^{-1} \eta(s_j, a_j)\right]^{1/2} = \frac{1}{\sqrt{N_{s_j, a_j} + \lambda}}. \tag{43}$$

Thus, in conclusion, we obtain that

$$
\begin{aligned}
r^{\mathrm{db}}(s_j, a_j) = I\left(W_t; [s_j, a_j, S_{j+1}] \mid \mathcal{D}_m\right) &\approx \sqrt{\frac{c}{2}} \left[\eta(s_j, a_j)^\top \Lambda_t^{-1} \eta(s_j, a_j)\right]^{1/2} \\
&= \sqrt{\frac{c}{2}} \frac{1}{\sqrt{N_{s_j, a_j} + \lambda}} = \beta_0 \cdot r_j^{\mathrm{count}},
\end{aligned}
\tag{44}
$$

where $c = |Z|$ is the same as the number of states $|\mathcal{S}|$ in tabular MDPs. Thus, we complete the proof of Theorem 2. The bonus become smaller when the corresponding state-actions are visited more frequent, which follows the principle of count-based exploration [9, 38]. $\qquad\square$

# D  Implementation Detail

Table 1: Experimental setup of PPO. The implementation of PPO is the same for all methods in our experiments.

| Hyperparameters | Value | Description |
|---|---|---|
| state space | $84 \times 84 \times 4$ | Stacking 4 recent frames as the input to network. |
| action repeat | 4 | Repeating each action 4 times. |
| actor-critic network | conv(32,8,4) conv(64,4,2) conv(64,3,1) dense$\{512, 512\}$ dense$|\mathcal{A}| + 1$ | Using convolution(channels, kernel size, stride) layers first, then feed into two fully-connected layers, each with 512 hidden units. The outputs of the policy and the value function are split in the last layer. The output of policy is a Softmax function. The output of the value function is a single linear unit. |
| entropy regularizer | $10^{-3}$ | The loss function of PPO includes an entropy regularizer of the policy to prevent premature convergence |
| $\gamma_{\mathrm{ppo}}$ | 0.99 | Discount factor of PPO |
| $\lambda_{\mathrm{ppo}}$ | 0.95 | GAE parameters of PPO |
| normalization | mean of 0, std of 1 | Normalizing the intrinsic reward and advantage function by following [39, 11]. The advantage estimations in a batch is normalized to achieve a mean of 0 and std of 1. The intrinsic rewards is smoothen exponentially by dividing a running estimate of std. |
| learning starts | 50000 | The agent takes random actions before learning starts. |
| replay buffer size | 1M | The number of recent transitions stored in the replay buffer. |
| training batches | 3 | The number of training batches after interacting an episode for all actors. |
| optimizer | Adam | Adam optimizer is used for training. Detailed parameters: $\beta_1 = 0.9$, $\beta_2 = 0.999$, $\epsilon_{\mathrm{ADAM}} = 10^{-7}$. |
| mini-batch size | 32 | The number of training cases for gradient decent each time. |
| learning rate | $10^{-4}$ | Learning rate for Adam optimizer. |
| max no-ops | 30 | Maximum number no-op actions before an episode starts. |
| actor number | 128 | Number of parallel actors to gather experiences. |
| $T$ | 128 | Episode length. |

Table 2: Experimental setup of DB

| Hyperparameters | Value | Description |
|---|---|---|
| network architecture | Appendix B | See Fig.2. |
| $\alpha_1$ | $\{0.1, 0.001\}$ | Factor for $I_{\mathrm{upper}}$. |
| $\alpha_2$ | 0.1 | Factor for $I_{\mathrm{pred}}$. |
| $\alpha_3$ | $\{0.1, 0.01\}$ | Factor for $I_{\mathrm{nce}}$. |
| $\tau$ | 0.999 | Factor for momentum update. |

Table 3: Comparison of model complexity. (1) ICM estimates the inverse dynamics for feature extraction with 2.21M parameters. ICM also includes a dynamics model with 2.65M parameters to generate intrinsic rewards. (2) Disagreement use a fixed CNN for feature extraction thus the trainable parameters is 0. Disagreement uses an ensemble of dynamics with total 26.47M parameters to estimate the uncertainty. (3) CB does not requires any additional parameters compared to the actor-critic network. (4) DB requires slightly more parameters than ICM in representation learning, while do not uses additional parameters in estimating the dynamics since the DB-bonus is directly derived from the information gain of latent representation.

| | Feature extractor | Dynamic model | Total |
|---|---|---|---|
| ICM | 2.21M | 2.65M | 4.86M |
| Disagreement | 0M | 26.47M | 26.47M |
| CB | 0M | 0M | 0M |
| **DB (ours)** | 5.15M | 0M | 5.15M |

# E Supplementary Experimental Results

## E.1 Random-Box Noise

We present the evaluation curves of Atari games with *random-box noise* in Fig. 7.

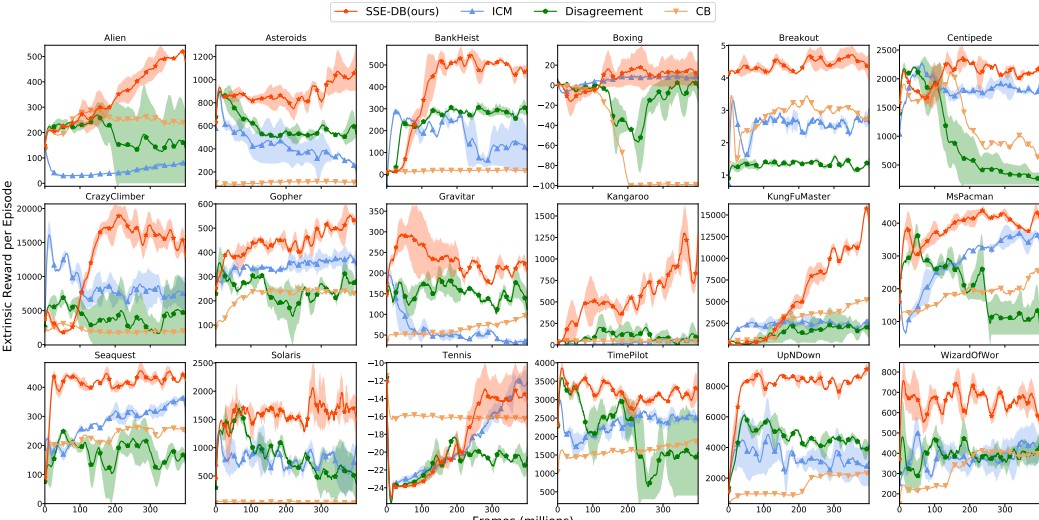

Figure 7: Evaluation curves in Atari games with *random-box noise*. SSE-DB outperforms all the baselines in 17 out of the 18 tasks. Comparing to standard Atari, we observe that the performance of SSE-DB with random-box noise is suboptimal in Breakout, Gopher, and WizardOfWar. In these tasks, dynamic-relevant information (e.g., the ball in Breakout, the tunnels in Gopher, and the worriors in WizardOfWar) are (partly) masked by random-boxes. Breakout is affected by random boxes significantly as the ball is too small to be distinguished from the noise. In addition, we observe that ICM is prone to random-box noise in most of the tasks. Disagreement also demonstrates decreased (e.g., CrazyClimber, Gopher, Kangaroo) or unstable (e.g., Alien, Centipede, TimePilot) performance in most of the tasks. A possible explanation is that the random-box noise affects the training of the dynamics models, thus bring adverse impact in estimating the Bayesian uncertainty through ensembles in Disagreement. In contrast, SSE-DB performs well in the presence of the random-box noise.

## E.2  Pixel Noise

We present the evaluation curves in Atari games with *pixel noise* in Fig. 8 and Fig. 9.

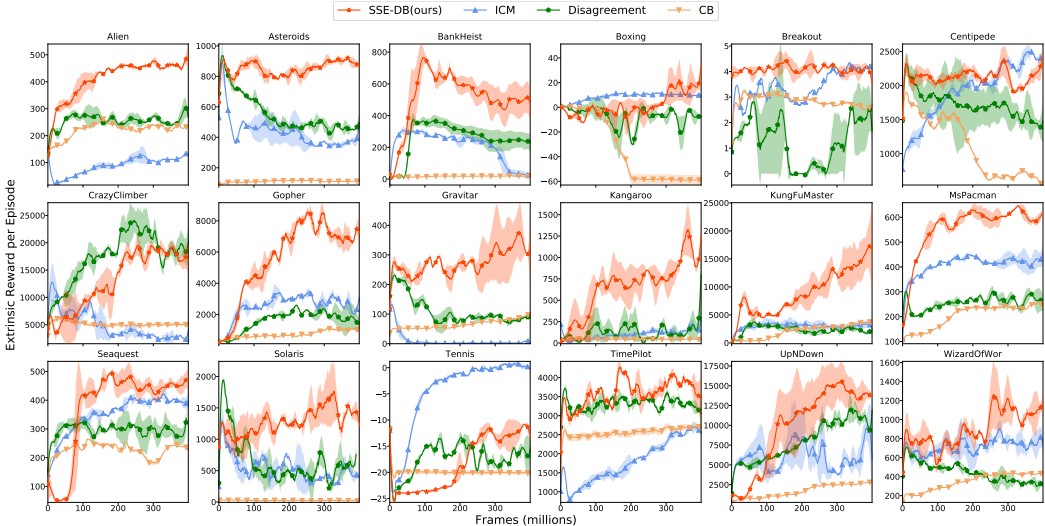

Figure 8: Evaluation curves in Atari games with *pixel noise*. SSE-DB outperforms all the baselines in 15 out of the 18 tasks. For games with the pixel noise, the performance of SSE-DB is similar to that for standard Atari games, expect for Breakout, in which the ball is too small to be distinguished from the pixel noise. We refer to Fig. 9 for a comparison of results with and without pixel noise.

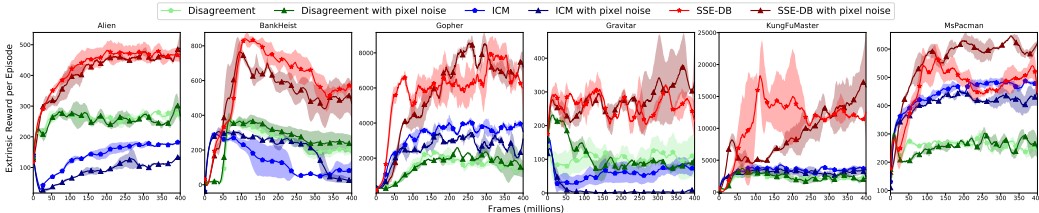

Figure 9: A comparison of results on selected Atari games with and without *pixel noise*. SSE-DB shows robustness to pixel noise since we discard the dynamics-irrelevant information. Nevertheless, we find that the adverse effect of introducing pixel noise is limited for exploration, both in SSE-DB and other baselines. Especially, in *Gravitar* and *MsPacman*, the performance of SSE-DB has sight improvement compared to that of standard Atari games. A possible explanation is that introducing the pixel noise leads to data augmentation for image inputs, thus bringing regularization for learning and enhance the generalization ability of the policy.

## E.3 Sticky Actions

Evaluation curves in Atari games with *sticky actions* is give in Fig. 10.

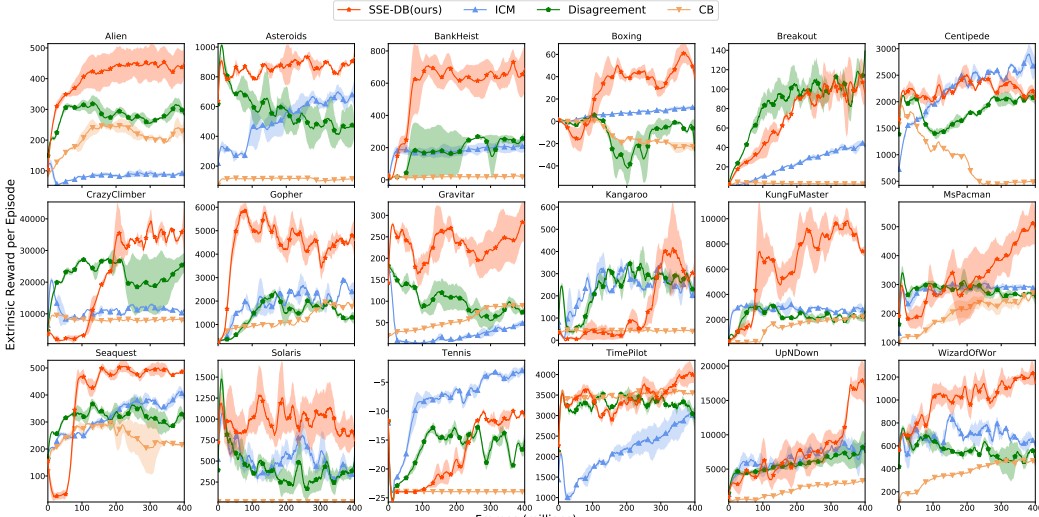

Figure 10: Evaluation curves in Atari games with *sticky actions*, which introduces additional stochasticity to the transition dynamics. As shown in figures, SSE-DB outperforms all baselines in 15 out of 18 games. We observe that in Breakout, MsPacman, and UpNDown, SSE-DB performs suboptimal in the early stage of training stage. Nevertheless, the performance gradually improves as the training evolves and reaches to the optimum eventually, which occurs since DB removes the noisy information for the state representation along the training process. We highlight that SSE-DB compresses the latent representation of the corresponding state-action pair, thus minimizing the effect of noisy actions. In addition, Disagreement method based on Bayesian uncertainty also demonstrates robustness to sticky actions, despite the fact that it needs more parameters than other methods according to Table 3.

### E.4 Ablation Study

In this section, we present several ablation studies to analyze the components of the DB model. We conduct experiments on different settings that remove different components in DB, namely,

- *No-Upper*, which is a variant of DB that removes $I_{\text{upper}}$ from $\mathcal{L}_{\text{DB}}$ (set $\alpha_1 = 0$) in (8);
- *No-Pred*, which is a variant of DB that removes $I_{\text{pred}}$ from $\mathcal{L}_{\text{DB}}$ (set $\alpha_2 = 0$);
- *No-NCE*, which is a variant of DB that removes $I_{\text{nce}}$ from $\mathcal{L}_{\text{DB}}$ (set $\alpha_3 = 0$); and
- *No-NCE-Momentum*, which is a variant of DB that removes $I_{\text{nce}}$ and utilizes the same encoder $f^S(;\theta)$ for successive observations $o_t$ and $o_{t+1}$, in contrast with the momentum observation encoding for $o_{t+1}$ of DB.

We conduct experiments on the Alien task with standard observation and random-Box noise. We illustrate the results in Fig. 11 and Fig. 12, respectively. We illustrate the performance comparison on extrinsic rewards in (a) of Fig. 11, and the change of $I_{\text{upper}}$, $I_{\text{pred}}$ $I_{\text{nce}}$ in (b), (c) and (d), respectively, of Fig. 11. We discuss the results in the sequel.

- *No-Upper* setting exhibits similar performance as SSE-DB in standard Atari. Without compressing the representation through minimizing $I_{\text{upper}}$, the latent $Z$ preserves more information and exhibits well exploration with DB-bonus. However, in random-box setting, the latent $Z$ contains distractors features thus bringing adverse effects in exploration. Interestingly, $I_{\text{upper}}$ first increases and then decrease without minimizing the $I_{\text{upper}}$ explicitly (see Fig. 11 (b) and Fig. 12 (b)). This phenomenon is reminiscent of previous studies of Information Bottleneck in Deep Learning [54, 48], suggesting that the neural network has the ability to actively compress the input for efficient representation. Nevertheless, according to our experiments, such compression is still not sufficient to handle the random-box noise in exploration. In such environments, the objective $I_{\text{upper}}$ needs to be minimized to discard dynamics-irrelevant features.

- *No-Pred* setting has significantly reduced performance and stability for both the standard observations and the observations with random-box noise, comparing with SSE-DB. According to our results, the contrastive estimation cannot replace the role of predictive objective in maximizing the mutual information of $Z_t$ and $S_{t+1}$. A possible explanation is that the contrastive loss is applied in a transformed space through projection heads, thus part of the dynamics-relevant information is encoded in the projection encoding but not the latent representation $Z$. However, our DB-bonus is defined in $Z$, thus do not capture the information encoded in projection heads. Besides, according to Fig. 11 (c) and Fig. 12 (c), the predictive objective does not increase through solely maximizing $I_{\text{nce}}$.

- *No-NCE* setting exhibits similar performance as DB in standard Atari. Since we use the momentum observation encoder to prevent collapsing solution, the lack of contrastive estimation does not bring significant performance loss. Meanwhile, according to Fig. 11 (d) and and Fig. 12 (d), we find that $I_{\text{nce}}$ improves slightly without being optimized explicitly, since the predictive objective helps improving $I_{\text{nce}}$ implicitly. Nevertheless, the lack of NCE-loss produces reduced performance with random-box noise. Our experiments show that using contrastive estimation leads to a stronger distillation for the dynamics-relevant feature of observations.

- *No-NCE-No-Momentum* setting leads to the collapsing solution in representation. According to Fig. 11 (c) and and Fig. 12 (c), the predictive objective converges very fast by encoding all observations to uninformative values. Such a trivial solution is undesirable as it does not capture any meaningful information. The performance of such setting decrease significantly comparing against SSE-DB.

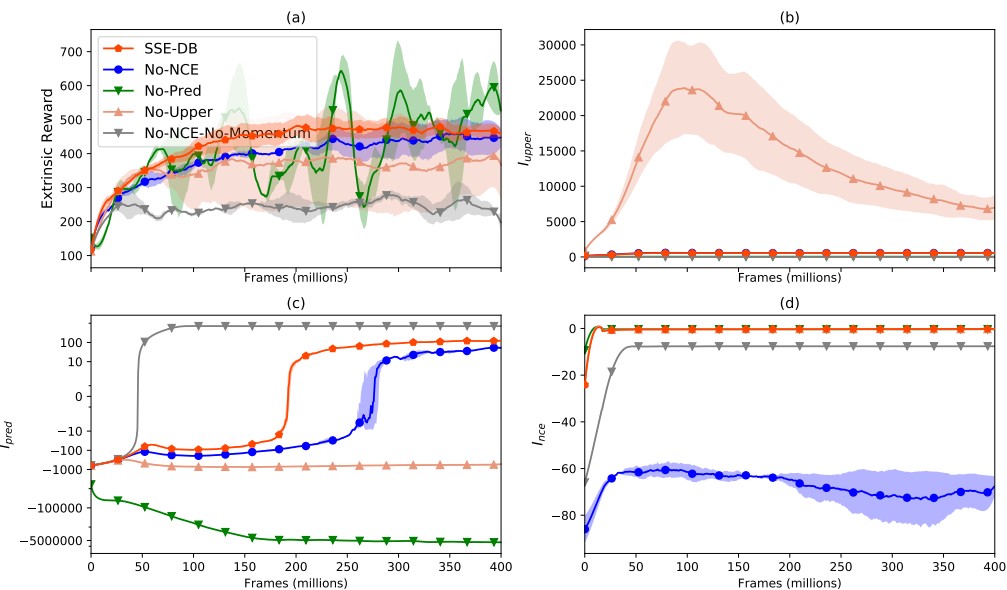

Figure 11: Ablation study in *Alien*. We measure (a) extrinsic rewards, (b) $I_{\text{upper}}$ that indicates the amount of information contained in the representation space, (c) the predictive objective $I_{\text{pred}}$, and (d) the contrastive estimation $I_{\text{nce}}$ for comparison.

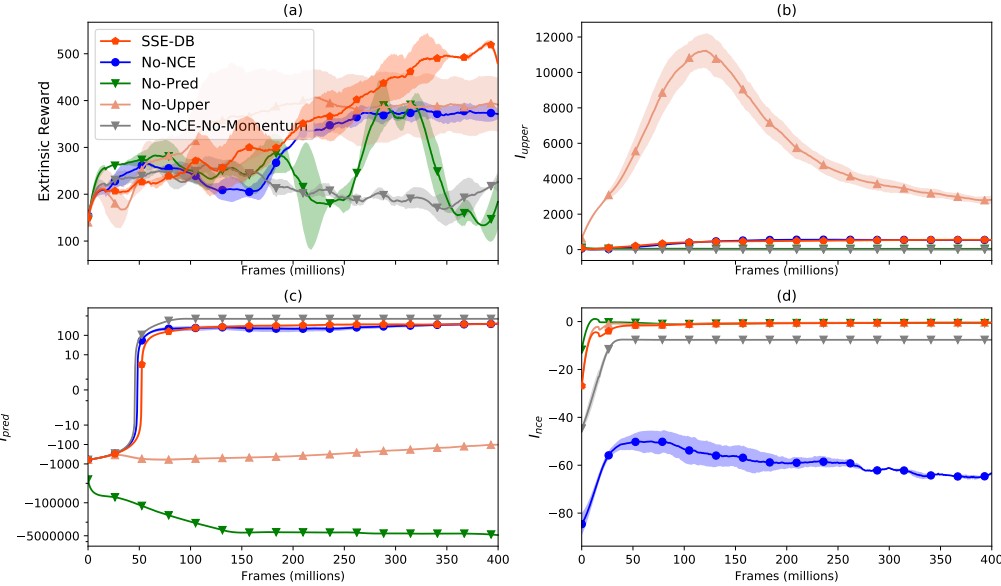

Figure 12: Ablation study in *Alien* with *random-box noise*. We measure (a) extrinsic rewards, (b) $I_{\text{upper}}$ that indicates the amount of information contained in the representation space, (c) the predictive objective $I_{\text{pred}}$, and (d) the contrastive estimation $I_{\text{nce}}$ for comparison.

### E.5  Visualizing DB-bonus

The proposed DB-bonus motivates the agent to explore states and actions that have high information gain to the representation. To further understand the DB-bonus, we provide visualization in two tasks to illustrate the effect of DB-bonuses. We choose two Atari games Breakout and Gopher, and visualize the DB-bonus in an episode based on a trained DB model.

#### E.5.1  Breakout

In Breakout, the agent uses walls and the paddle to rebound the ball against the bricks and eliminate them. We use a trained SSE-DB agent to interact with the environment for an episode in Breakout. The whole episode contains 1942 steps, and we subsample them every 4 steps for visualization. The curve in Fig. 13 shows the UCB-bonus in 481 sampled steps.

We select 16 spikes of the DB-bonus on the trajectory and then visualize their corresponding observations. From the results, we find that the spikes typically correspond to some critical observations, including eliminating bricks (e.g., frames 1, 3, and 5-8), rebounding the ball (e.g., frames 2 and 4), digging a tunnel (e.g., frames 9-12), and throwing the ball onto the top of bricks (e.g., frames 13-16). These examples demonstrate that the DB-bonus indeed encourages the agent to explore many crucial states, even without knowing the extrinsic rewards. The DB-bonus encourages the agent to explore the potentially informative and novel state-action pairs to get high rewards. We also record 15 frames after each spike for further visualization. The video is available at https://www.dropbox.com/sh/gw7m38o29dfl9zx/AACF1AogB93spuD_Vsk_lsOBa?dl=0.

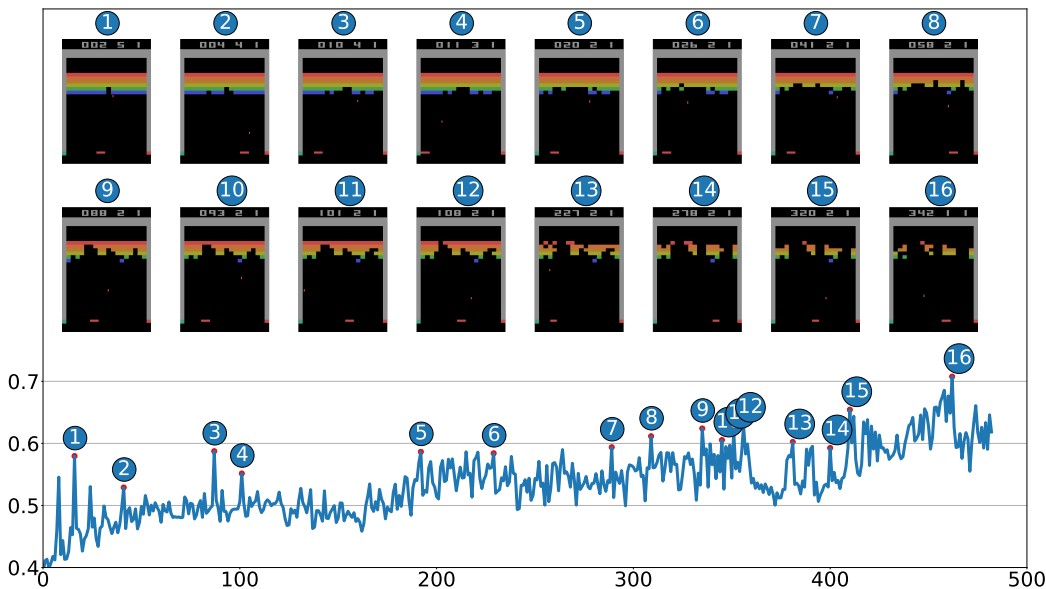

Figure 13: Visualization of DB-bonus for an episode in Breakout. The curve corresponds to the DB-rewards of a trajectory. The numbers from 1 to 16 corresponds to the selected spikes, and the images are the corresponding observations.

#### E.5.2  Gopher

Gopher is a popular Atari game. In this game, the gopher tunnels left, right, and up to the surface. When the gopher makes a hole to the surface, he will attempt to steal a carrot. If the holes have been tunneled to the surface, the farmer (i.e., the agent) can hit the gopher to send him back underground or fill in the holes to prevent him from reaching the surface. Rewards are given when the agent hits the holes and gopher. SSE-DB performs well in this task. To illustrate how DB-bonus works, we use an SSE-DB agent to play this game for an episode and records the DB-bonus in all 4501 steps. Fig. 14 shows the DB-bonus and the corresponding frames in 16 chosen spikes in 1125 subsampled steps.

We find almost all spikes (i.e., frames 1-3, 5-13, 15-16) of DB-bonus correspond to scenarios that the gopher makes a hole to the surface, which is rarely occurs and signifies that the gopher will have a chance to eat carrots. Also, these scenarios are crucial for the farmer to get rewards since the farmer can hit the gopher and holes to prevent the carrots from being eaten. The DB-bonus encourages the farmer to learn skills to move fast and hit the holes in the surface to obtain high scores. In addition, the gopher moves underground in frames 4 and 14, and the farmer mends holes in the surface. The transitions with high DB-bonuses make the agent explore the environment efficiently. The video of spikes is available at `https://www.dropbox.com/sh/gw7m38o29dfl9zx/AACF1AogB93spuD_Vsk_lsOBa?dl=0`.

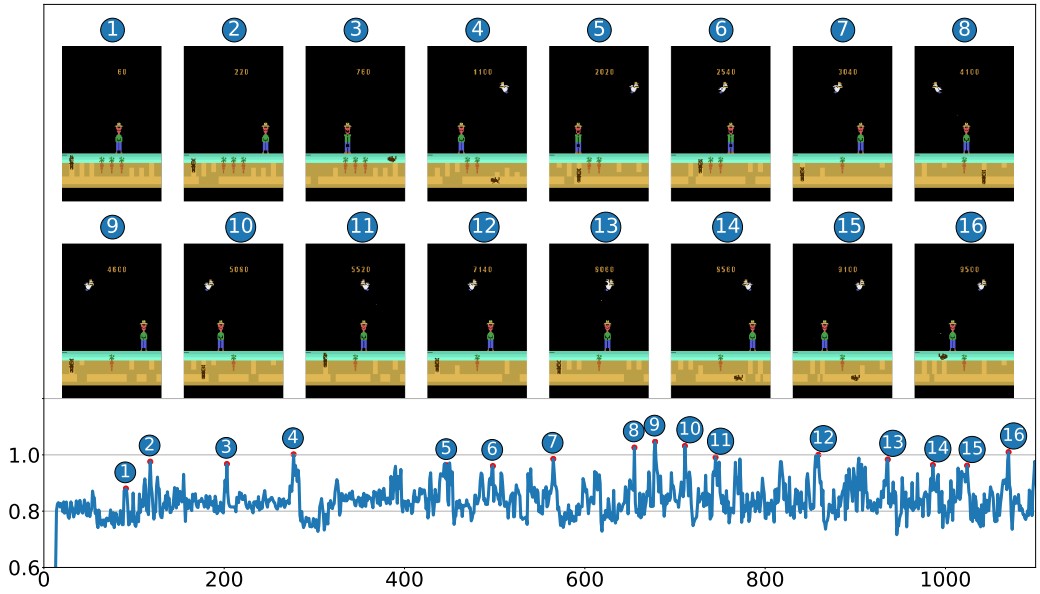

Figure 14: Visualization of DB-bonus for an episode in *Gopher*.

### E.6  Montezuma's Revenge

Several exploration methods demonstrate superior performance on Montezuma's Revenge [51]. However, these methods all use intrinsic rewards along with extrinsic rewards from the environment in training. In the self-supervised exploration setting where the training hinges solely on the intrinsic rewards, we find that SSE-DB and all the other baselines scores zero in this task.

Nevertheless, we observe that the agent trained with DB-bonus can pass half of the first room in Montezuma's Revenge. We conduct t-SNE [31] visualization to illustrate the learned representation of DB. Form Fig. 15(a), we find that the latent representations are well aligned in several clusters, which corresponds to stepping down the ladder, jumping to the pillar, and escaping an enemy, respectively. We also visualize the raw states from the same episode with t-SNE in Fig. 15(b). In contrast, we do not find any meaningful clusters form the visualization of raw states. DB enables to capture certain aspects of such a hard task.

We give a video of the trained policy of DB [1]. From the video, we find that the self-supervised agent with DB-bonus can learn some skills, including stepping down the ladder, jumping to the pillar, and trying to escape the enemy. Nevertheless, as the learning curve suggests, such learned skills along are insufficient for obtaining scores.

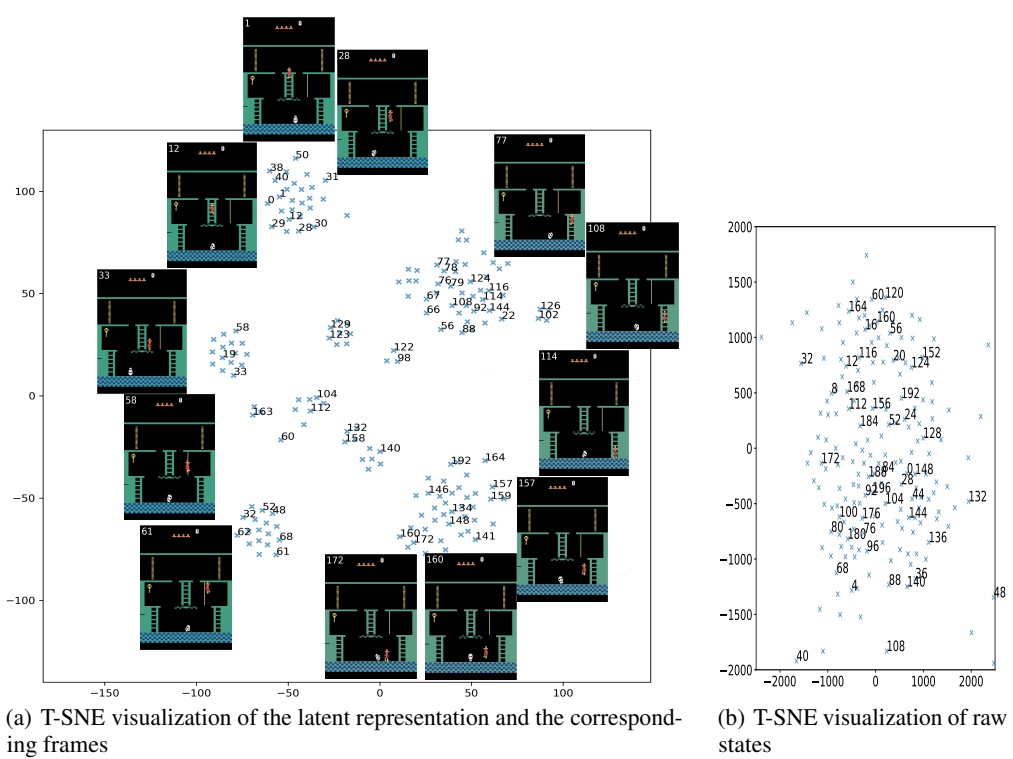

(a) T-SNE visualization of the latent representation and the corresponding frames

(b) T-SNE visualization of raw states

Figure 15: (a) Visualization for 200 observations of Montezuma's Revenge in an episode. We visualize the latent representations of DB in 2 dimensions with t-SNE [31]. Numbers on top-left of game frames correspond to numbers of representations in lower-dimensional space. (b) Visualization for raw states ($84 \times 84 \times 4$ for each one) in the same episode.

---

[1] https://www.dropbox.com/s/boijqmt66mgnj17/montezuma-DB.mp4?dl=0

## E.7 Comparison with Entropy-Based Exploration

There exist several methods that perform entropy-based exploration for unsupervised representation learning, and then use this representation for downstream task adaptation, including VISR [19], APT [29], APS [28], RE3 [47] and Proto [59]. The entropy-based methods use $k$-nearest neighbor state entropy estimator to estimate the entropy of state $\mathcal{H}(s)$ and then use it as intrinsic rewards.

In this section, we focus on the unsupervised exploration stage and compare SSE-DB with entropy-based exploration methods. Since APT and APS do not release code and ProtoRL conducts experiments in DeepMind control rather than Atari, we conduct experiments with RE3 algorithm. Since RE3 uses Rainbow as the basic algorithm, and uses both the extrinsic and intrinsic rewards in training, we re-implement the RE3 bonus in our codebase to evaluate its performance in a self-supervised setting with noisy environments. As shown in Fig. 16, RE3 performs reasonably in standard Atari games. However, the performance decreased significantly in the Random-Box Atari environments. A possible reason is that the entropy of the state increases significantly if we inject noises. Hence, exploration is misled by the noises in these environments. Nevertheless, as shown in Fig. 17, the entropy-based methods are robust to sticky actions since they use the entropy of states in exploration, without considering the entropy of actions.

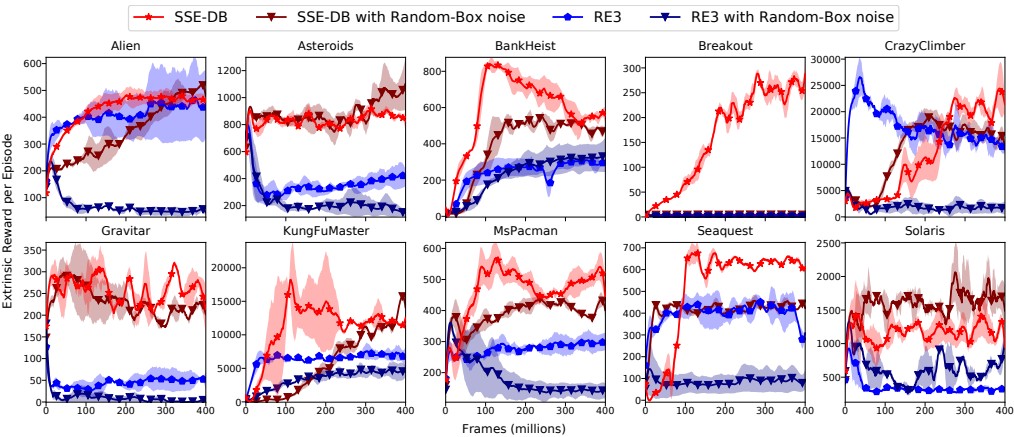

Figure 16: A comparison of results on selected Atari games with and without Random-Box noise. SSE-DB shows robustness to Random-Box noise while RE3 is sensitive to the noises.

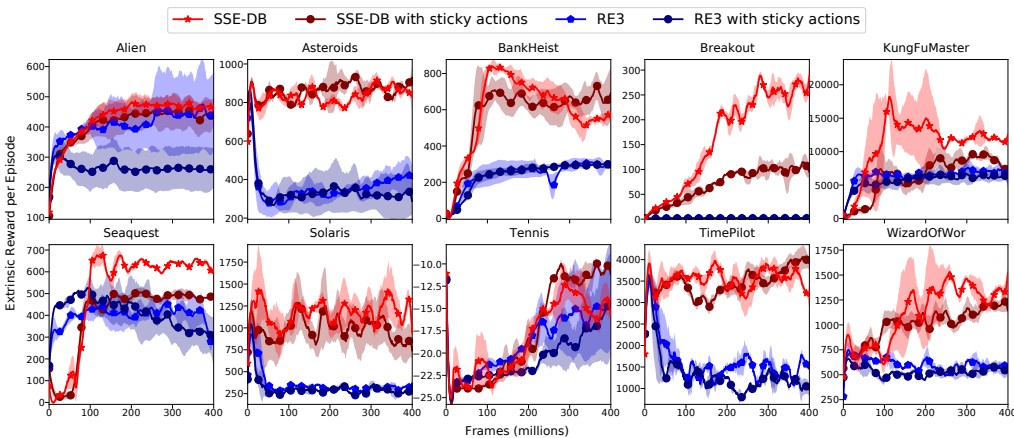

Figure 17: A comparison of results on selected Atari games with and without sticky actions. Both SSE-DB and RE3 show robustness to sticky actions.