# OpenReview forum: "Dynamic Bottleneck for Robust Self-Supervised Exploration"
_NeurIPS.cc/2021/Conference — NeurIPS 2021 Poster_

### Official Review · Reviewer_sHR8 · 2021-07-12

**Rating:** 6
**Confidence:** 3

**Summary:**

The authors introduce the dynamics bottleneck which uses the information bottleneck to discover the novelty of state-action pairs to introduce a reward bonus, used for exploration in RL tasks. The authors evaluate their method on top of PPO in the Atari domain.

**Limitations And Societal Impact:**

This is addressed. Although I would encourage the authors to move their comments from the checklist to either the main paper or the supplementary material.

**Main Review:**

Strengths:

- Overall, the empirical results look good compared to related approaches.
- Theoretical results drawing the connection between the proposed approach and different bonuses in the exploration literature shows this approach is reasonable.
- I thought the supplementary was very complete, reproducibility of the paper is very high. Supplementary includes an ablation study.

Weaknesses:

- For a paper that aims to tackle the task of exploration, I thought the choice of environments was a bit lacking in specifically hard exploration tasks such as Montezuma's revenge or modified sparse reward environments. For example [1] lists (Gravitar, Montezuma’s Revenge, Pitfall!, Private Eye, Solaris, and
Venture) as hard exploration tasks. Out of these 6, the 2 in tested in this paper (Solaris and Gravitar) have mostly flat learning curves. Compared to the learning curves from Dopamine [2], the performance of SSE-DB doesn't seem to be much better than the randomly initialized policy on these tasks. ~~According to the PPO paper [3], on Gravitar (Solaris results were not included in this paper), vanilla PPO can achieve a performance of 500-750+ after 40M frames, which is than SSE-DB in fewer frames.~~ (Edit: not a valid concern)
- Clarity on novelty. One thing that wasn't too clear to me after the reading the paper is what of the approach was novel and what was based on prior work. For example 3.1 and 3.3 are also described in the CB paper [4] (with references to prior work), so it isn't clear to me if there is novelty here, or if the novelty was limited to viewing the information bottleneck over the dynamics. A similar concern for 3.2 as (as mentioned in the related work) there has been literature on contrastive learning in RL.

Minor:
- In the section 5.2 on the experiment in the presence of noise, is SSE-DB was only using the learned reward bonus, or is the learned representation used in some more meaningful way?

References:
- [1] Burda, Yuri, et al. "Exploration by random network distillation." 2018.
- [2] Castro, Pablo Samuel, et al. "Dopamine: A research framework for deep reinforcement learning." 2018.
- [3] Schulman, John, et al. "Proximal policy optimization algorithms." 2017.
- [4] Kim, Youngjin, et al. "Curiosity-bottleneck: Exploration by distilling task-specific novelty." 2019.

**Time Spent Reviewing:**

5

---

> ### Author Response · Authors · 2021-08-10
> **Response to reviewer sHR8**
>
>
> We thank the reviewer for the valuable comments and time dedicated to evaluating our work.
>
> 1. In many previous exploration methods, the reward $r$ used for training is a linear combination of extrinsic reward $r^e$ and intrinsic reward $r^i$, i.e., $r = r^e + \alpha r^i$, where $\alpha$ is a tuning parameter. In contrast, in our paper, we consider the self-supervised exploration setting (or pure exploration), which only uses intrinsic reward in training, i.e., $r=r^i$. Such setting is significantly more difficult since the intrinsic rewards become the only driving force of the entire learning process. We have to clarify that both Dopamine and PPO use extrinsic rewards in training, which is the reason that their performances are better than ours.
>
>     Actually, We find that without extrinsic rewards, most exploration methods perform poorly in Montezuma's Revenge. In recent empirical work, Taiga et al [1] extensively compares exploration methods on Montezuma's Revenge. The results show that although some methods achieve good performance in Montezuma's Revenge (e.g. RND obtains approximately 8150 points with extrinsic rewards), they do not perform well on other Atari games (e.g, RND performs similar to $\epsilon$-greedy in other Atari games). Based on this prior experience, we add an experiment to train DB in Montezuma's Revenge with extrinsic rewards being available, which achieves around 4120 points. We also provide a visualization analysis in Appendix E.6, which shows that the latent representation in DB is well aligned in several clusters. Although the obtained score may be suboptimal comparing with the best baselines in solving Montezuma's Revenge, we would like to clarify that our proposed SSE-DB is not specifically targeted to solve such a task. Moreover, DB focuses on robust exploration and has more advantages in noisy environments according to the empirical results in Section 5.2 and Appendix E. We will provide all the above details and discussions in the revision.
>
> 2. Our method is related to CB [3] in the sense that both CB and DB follow the principle of Information Bottleneck (IB). Nevertheless, DB is different from CB in two directions: (1) CB learns to extract task-relevant information, where `task' is identified by environmental rewards. However, when the environmental rewards are sparse or entirely unavailable (which is studied in our paper), CB does not work well, which is also verified in our experiments. In contrast, DB learns the representation without the requirement of environmental rewards. (2) CB uses the compressiveness of observation with respect to the Q-network as the intrinsic rewards, which is not theoretically grounded. In contrast, we provide theoretical connections between DB-bonus and the provably efficient UCB-bonuses.
>
>     The contrastive learning methods in RL typically use data augmentation to construct positive and negative samples [4], while we use dynamics to obtain positive and negative samples. The positive pairs are sampled from the joint transition distribution, while the negative pairs are sampled from the marginal distribution. In addition, the previous methods often use contrastive learning as auxiliary losses in policy optimization, while we focus on robust exploration and use contrastive learning as a part of the IB objective to generate bonuses for exploration.
>
> Minor:
>
> Yes, DB is only used to provide DB-bonus for exploration in our experiments. For a fair comparision to baselines, we do not change the representation in policy optimization. Different baselines use the same policy optimization method but with different types of bonus for exploration. Applying DB representation in policy optimization (like in [4]) for sample-efficient learning is an interesting direction, and we would like to investigate that in future.
>
> References
>
> [1] Taiga, Adrien Ali, et al. "On bonus based exploration methods in the arcade learning environment." International Conference on Learning Representations. 2020.
>
> [2] Burda, Yuri, et al. "Exploration by random network distillation." International Conference on Learning Representations. 2019.
>
> [3] Kim, Youngjin, et al. "Curiosity-bottleneck: Exploration by distilling task-specific novelty." International Conference on Machine Learning. 2019.
>
> [4] Laskin M, Srinivas A, Abbeel P. "Curl: Contrastive unsupervised representations for reinforcement learning." InInternational Conference on Machine Learning. 2020

---

> > ### Comment · Reviewer_sHR8 · 2021-08-19
> > **Response to Authors**
> >
> > Thank you for the response and clarifications.
> >
> > You are correct about the role of extrinsic rewards in PPO, although I stand by the fact that the method doesn’t achieve much performance on these hard exploration tasks, this is likely less of a concern than I had originally thought.

---

> > > ### Author Response · Authors · 2021-08-24
> > > **Follow-up response to reviewer sHR8**
> > >
> > > Thanks for your valuable feedback. Our submission includes results of three hard-exploration games (Montezuma's Revenge, Solaris, and Gravitar) among Atari games. We add experiments of the other three tasks (Pitfall, PrivateEye, and Venture). The results are given in [1]. Our method outperforms other baselines in Pitfall and Venture, and performs similar to other baselines in PrivateEye. We remark that the scores achieved by all self-supervised baselines are relatively low, which suggests that relying solely on intrinsic rewards is insufficient to solve these tasks.
> > >
> > > [1] See experiments at the end of Readme at: https://github.com/review-anon/DB

---

### Official Review · Reviewer_LMHD · 2021-07-16

**Rating:** 6
**Confidence:** 4

**Summary:**

This paper propose an exploration method for reinforcement learning by introducing a dynamics bottleneck (DB) model that captures dynamics-revelant information from the states. To this end, they propose to maximize the mutual information between $Z_{t}$ and $S_{t+1}$ but minimize the mutual information between $Z_{t}$ and $(S_{t}, a_{t})$ to compress representation. Intrinsic reward based on learned DB-model. The proposed method is evaluated on tasks from Atari Learning Environment and corrupted Atari tasks.

**Limitations And Societal Impact:**

The draft does not address limitations and potential negative societal impact in the main text. They should include the relevant discussion in the main text following the suggestion of NeurIPS conference.

**Main Review:**

Proposed method is not surprising, but seems novel and reasonable, writing is clear, and its effectiveness is supported by extensive experimental results. I like the paper in general, but I have some minor concerns that could be addressed easily, and i'm willing to increase the score when the concerns below are resolved:

- Positioning of the proposed method in the paragraph 2 of introduction is a bit questionable. For example, representations learned by inverse dynamics model you mentioned as 'curiosity-driven explorations' should capture controllable features (relevant discussions are in [Pathak'17; Badia'20]), which in principle should not be affected by dynamics irrelevant objects like birds. While this aspect ​of capturing dynamics-relevant information is the very important characteristic of the proposed method and the method could be very good at doing it, it could not be its uniqueness compared to other related works. Could you elaborate on this and provide more intuitive explanation and supporting evidence why the proposed method should be better than the previous works?

- While I appreciate that experimental results are extensive, but important and more recent baselines for self-supervised exploration are missing, like VISR [Hansen'20], APT [Liu'21a], Plan2Explore [Sekar'20], which were all available at the time of submission (via conference, arXiv, OpenReview). As a note, much more recent works like APS [Liu'21b], RE3 [Seo'21], ProtoRL [Denis'21] are also relevant.

- How are hyperparameters ($\alpha_1, \alpha_2, \alpha_3$) are selected? Is the proposed method sensitive to the choice of these hyperparameters?

Minor
- line 38: discards -> discard
- line 137: plays
- line 292: observer - > observe

[Pathak'17] Curiosity-driven Exploration by Self-supervised Prediction

[Sekar'20] Planning to Explore via Self-Supervised World Models

[Hansen'20]  Fast Task Inference with Variational Intrinsic Successor Features

[Liu'21a] Behavior From the Void: Unsupervised Active Pre-Training

[Liu'21b] APS: Active Pretraining with Successor Features

[Seo'21] State Entropy Maximization with Random Encoders for Efficient Exploration

[Yarats'21] Reinforcement Learning with Prototypical Representations

**Time Spent Reviewing:**

8 hours

---

> ### Author Response · Authors · 2021-08-10
> **Response to reviewer LMHD**
>
>
> We thank the reviewer for the valuable comments and time dedicated to evaluating our work.
>
> 1. We provide an intuitive example to explain why learning inverse dynamics cannot filter out the white-noise in representation. Let $c$ be the noise and $([s,c],a, s')$ be a transition with noise in state. Let $z = [\phi(s), \psi(c), \phi(s')]$ be the embedded representation. The issue is that learning inverse dynamics does not preclude learning such a representation $z$ that incorporates the irrelevant noise. The reason is that in the prediction of action, the output layer could set the connection with $\psi(c)$ to be zero, which can also recover the action $a$ accurately without being affected by the noise. Thus, learning inverse dynamics does not guarantee the removal of noise in representation. The representation that incorporates noise would thus affect the exploration since the bonus is calculated based on the representation.
>
>     In contrast, DB explicitly compresses the representation by minimizing the MI between $z$ and the input $([s,c],a)$ where $c$ denotes noise. Through maximizing $I(Z; S')$ and minimizing $I(Z; ([S,C],A))$, agent with DB representation can learn to discard the information of noise $c$ in $z$ since $c$ does not carry information of $s'$.
>
> 2. We thank the reviewer for the suggested related works. We have carefully read these approaches and compare them to our work in the sequel.
>
> - Plan2Explore [Sekar'20] is a model-based method. Plan2Explore learns a world model to plan and conduct exploration by novelty in transition. The novelty of transition is measured by a similar mechanism to Disagreement [1], which maintains several dynamics models and uses the disagreement of these models as the intrinsic rewards. Disagreement [1] has been used as a baseline in our experiment in a model-free setting, and DB outperforms this method in most of the tasks.
>
> - VISR [Hansen'20] contains two stages, namely, unsupervised feature learning and task adaptation. In the sequel, we focus on the unsupervised learning stage. According to Table 1 in the VISR paper, VISR performs similarly to ICM [2] in the unsupervised stage, which is a baseline compared in our experiments. In addition, the authors comment in their paper [Hansen'20] that, *"it is clear that while VISR is not particularly outstanding in the unsupervised regime, when allowed 100k steps of RL it can vastly outperform these existing unsupervised methods on all criteria."* Thus, the focus of VISR is in fact the fast adaptation to tasks, which is not the focus of our work. We adopt some results from Table 1 of VISR.
>
>     |  methods   | median score  | mean score |
>     |  ----  | ----  | ----  |
>     | ICM  | 8.46 | 24.51 |
>     | VISR  | 4.04 | 58.47 |
>
> - APT [Liu'21a] and APS [Liu'21b] are similar works and use unsupervised skill discovery for fast task adaptation. In the unsupervised stage, they use the k-nearest neighbor state entropy estimator to estimate the entropy of state $H(s)$, and then use it as the intrinsic reward. Since APT and APS do not release code and the similar intrinsic rewards are also used in RE3 [Seo'21] and ProtoRL [Denis'21], we conduct experiments based on RE3 [Seo'21] and ProtoRL [Denis'21] for the comparison.
>
> - RE3 [Seo'21] uses a random encoder to get the state representation and uses the k-nearest neighbor state entropy estimator to estimate the entropy of state $H(s)$, which is the intrinsic reward in exploration. Since RE3 uses Rainbow as the basic algorithm, and uses both the extrinsic and intrinsic rewards in training, we re-implement the RE3 bonus in our codebase to evaluate its performance in a self-supervised setting with noisy environments. We release the experiment results and implementations at an anonymous repository [3]. The results show RE3 performs reasonably in *standard* Atari games. However, the performance decreased significantly in the *noisy* Atari environments. A possible reason is that the entropy of the state increases significantly if we inject noises. Hence, exploration is misled by the noises in these environments. Due to short response period, we are only able to conduct experiments in 5 games. We will provide full experiments on the remaining tasks in our revision.
>
> - ProtoRL [Denis'21] learns a prototypical representation and also uses state entropy as the intrinsic reward. This method includes an unsupervised stage to learn representation for downstream tasks. Since ProtoRL conducts experiments in DM control rather than Atari, it is hard to perform a direct comparison between ProtoRL and our SSE-DB. Alternatively, we construct noisy environments in DM control with the random-box and pixel noise to evaluate its robustness to noises. The code and results are released at an anonymous repository [4]. We find that ProtoRL is sensitive to random-box noise and the performance decreases significantly.
>
> 3. We did provide ablation study for the hyperparameters $\alpha_1,\alpha_2$ and $\alpha_3$ in the supplementary material. Please refer to Appendix E.4 where we set $\alpha_1,\alpha_2$ and $\alpha_3$ to zero, respectively, to compare the performances. We add additional ablation experiments in task `Alien' with random-box noise, and the results are given in [5]. In this task, the setting of $\alpha_1=\alpha_2=0.1,\alpha_3=0.01$ performs the best. We set all $\alpha=0.1$ in experiments by searching over 3 tasks (Alien, MsPacman and Seaquest). We found that setting $\alpha$ between $0.01$ to $0.1$ performs well in general.
>
> 4. The potential societal impact has been discussed in the checklist. We will move it to the main paper.
>
> [1] Pathak, Deepak, et al. "Self-supervised exploration via disagreement." International conference on machine learning. 2019.
>
> [2] Pathak, Deepak, et al. "Curiosity-driven exploration by self-supervised prediction." International conference on machine learning. 2017.
>
> [3] See experiments of RE3 with white-noise at: https://github.com/review-anon/RE3-White-Noise
>
> [4] See experiments of ProtoRL with white-noise at: https://github.com/review-anon/Proto-White-Noise
>
> [5] See ablation of $\alpha$ at: https://www.dropbox.com/s/1c3ux2na41wjefq/ablation_alpha.pdf?dl=0

---

> > ### Comment · Reviewer_LMHD · 2021-08-16
> > **Thanks for the response**
> >
> > Thank you very much for the response. I appreciate the time and effort of authors for providing intuitive explanation and experimental results.  And sorry for not clarifying this more: I think comparison with recent works (APS, RE3, ProtoRL) is not strictly required and just wrote them as a note, and appreciate for providing additional results. Incorporating the explanation on the difference to previous approaches and related works on self-supervised training would be helpful for improving the clarity of the draft. I will raise the score as most of my concerns are addressed.
> >
> > However I leave some remaining comments :
> >
> > - I'm not totally sure with the response that Disagreement is similar to Plan2Explore, as there is a critical difference between them in that Plan2Explore seeks the future novel states but Disagreement utilize the novelty of previously visited states (retrospective). However, I agree that both methods should be similarly sensitive to the noise hence it's not a critical concern for me.
> >
> > - I agree with Reviewer sHR8 and evaluating the method on more 'hard exploration' tasks should be helpful for evaluating the effectiveness of the proposed method.

---

> > > ### Author Response · Authors · 2021-08-24
> > > **Follow-up response to reviewer LMHD**
> > >
> > > Thanks for your valuable feedback.
> > >
> > > 1. We agree that Plan2Explore and Disagreement are different methods. Plan2Explore combines the environment model based on PlaNet [1] and the exploration bonus based on Disagreement [2], which allows the policy to learn from imagined states to seek future novelty. Nevertheless, we remark that such a model-based RL principle can also combine with other exploration bonuses (e.g., ICM, CB, and our DB) that are originally proposed in a retrospective manner (i.e., proposed for model-free RL). Such intrinsic-oriented methods have independent modules to generate bonuses, which can be used in both the model-based and the model-free methods. Thus, as far as we concern, a more important contribution of Plan2Explore is proposing a new principle for utilizing bonuses instead of introducing a new exploration bonus.
> > >
> > > 2. Thanks for the suggestion on additional experiments on hard exploration tasks. In the sequel, we briefly explain our intuition behind calibrating the exploration power of algorithms with the self-supervised RL problem set-up.
> > >
> > >     - We remark that a major reason for the hard exploration games to be *hard* is that their rewards are sparse. For example, in Montezuma's Revenge, the reward is sparse since it takes a long horizon for the agent to collect the key (+100) and open the door (+300) to get rewards. Nevertheless, if we manually assign additional rewards for critical scenarios (e.g., climbing down the ladder, avoiding the ghost, collecting the key, climbing up the ladder, and opening the door), then the task is no longer *hard* in terms of exploration [3]. Conversely, if we remove the extrinsic rewards for *easy* exploration games (e.g., the reward of hitting the box in Breakout), then the games after removing rewards become *hard* in terms of exploration. In our paper, we consider the self-supervised setting where all extrinsic rewards are removed. Thus, all tasks become harder in terms of exploration.
> > >
> > >     - Our submission includes results of three *hard* exploration games (Montezuma's Revenge, Solaris, and Gravitar) among Atari games. Following the suggestion of adding more hard exploration tasks, we add experiments of the other three *hard* exploration tasks (Pitfall, PrivateEye, and Venture) in our revision. The results are given in [4]. Our method outperforms other baselines in Pitfall and Venture, and performs similar to other baselines in PrivateEye. We remark that the scores achieved by all self-supervised baselines are relatively low, which suggests that relying solely on intrinsic rewards is insufficient to solve these tasks.
> > >
> > > [1] Hafner, D., et al. Learning latent dynamics for planning from pixels. ICML, 2019
> > >
> > > [2] Pathak, Deepak, et al. "Curiosity-driven exploration by self-supervised prediction." ICML. 2017.
> > >
> > > [3] Aytar, Yusuf, et al. "Playing hard exploration games by watching YouTube." NeurIPS. 2018.
> > >
> > > [4] See experiments at the end of Readme at: https://github.com/review-anon/DB

---

> > > > ### Comment · Reviewer_LMHD · 2021-08-25
> > > > **Thanks for the response**
> > > >
> > > > - Thanks again for the response. I agree with your point that reward-free environments are also 'hard' exploration games even if they are relatively easy tasks when external rewards are available. However, evaluation on standard 'hard' exploration games are still important as you're evaluating the methods using external rewards, which is sparse in the hard games.
> > > >
> > > > - In that sense, based upon (a) your response 'We remark that the scores achieved by all self-supervised baselines are relatively low, which suggests that relying solely on intrinsic rewards is insufficient to solve these tasks' and (b) the Reviewer sHR8's comment 'the performance of SSE-DB doesn't seem to be much better than the randomly initialized policy on these tasks', I'm not sure claiming that 'Our method outperforms other baselines in Pitfall and Venture, and performs similar to other baselines in PrivateEye' does have any meaning, as it seems like all methods fail to learn any meaningful behavior or skills even on these standard 'hard' Atari games without noise (e.g., zero performance on PrivateEye and Venture), and some methods unfortunately collapse to even worse policy than a random policy. (I used baseline [plots](https://google.github.io/dopamine/baselines/plots.html) from Dopamine for this response. Please correct me if i'm wrong here for interpreting the results.)
> > > >
> > > > - Hence, I think including the 'random policy' curves to the experimental results could be helpful here; this helps for making a conclusion whether the exploration methods can learn meaningful behavior or not. I think the method is still empirically good even if they are not perfectly working on hard exploration games, but making a tone bit down and explicitly stating the proposed method not working well on such environments (of course all other baselines also fail here) as limitation of the paper could make the contribution and limitation of the paper clear.
> > > >
> > > > - Sorry, where is the performance curve for Montezuma?

---

> > > > > ### Author Response · Authors · 2021-08-25
> > > > > **Follow-up response to reviewer LMHD**
> > > > >
> > > > > We appreciate the suggestions about the limitation of our work and agree that they are necessary for our work. Indeed, since pure exploration without extrinsic rewards is very difficult in most tasks, a random baseline is required to show whether the exploration methods learn meaningful behaviors. We adopt the random score from Table 2 of DQN [1] and Table 2 of Dueling DQN [2] to obtain the comparison. The experiment results with a random baseline are given in [3]. In hard exploration games such as PrivateEye, Solaris, and Montezuma's Revenge, both our method and the baselines cannot outperform random policy. Also, in Centipede and TimePilot, our method obtains similar scores to random policy. We will highlight the limitations of our algorithm in our revision.
> > > > >
> > > > > The results of Montezuma's Revenge are described in Appendix E.6 in our paper. We describe the result in text and do not give the learning curve. We have added the result in [3], showing that it scores zero in the training process for all methods. We give a video of trained policy in [4]. From the video, we find that the self-supervised agent with DB-bonus can learn some skills, including stepping down the ladder, jumping to the pillar, and trying to escape the enemy. Nevertheless, as the learning curve suggests, such learned skills along are insufficient for obtaining scores.
> > > > >
> > > > > [1] Mnih, Volodymyr, et al. "Human-level control through deep reinforcement learning." nature 518.7540 (2015): 529-533.
> > > > >
> > > > > [2] Wang, Ziyu, et al. "Dueling network architectures for deep reinforcement learning." ICML 2016
> > > > >
> > > > > [3] See experiment results with a random-policy baseline at: https://github.com/review-anon/DB
> > > > >
> > > > > [4] See video of Montezuma's Revenge with DB bonus at: https://www.dropbox.com/s/boijqmt66mgnj17/montezuma-DB.mp4?dl=0

---

> > > > ### Author Response · Authors · 2021-08-25
> > > > **Follow-up response to reviewer LMHD**
> > > >
> > > > We appreciate the suggestions about the limitation of our work and agree that they are necessary for our work. Indeed, since pure exploration without extrinsic rewards is very difficult in most tasks, a random baseline is required to show whether the exploration methods learn meaningful behaviors. We adopt the random score from Table 2 of DQN [1] and Table 2 of Dueling DQN [2] to obtain the comparison. The experiment results with a random baseline are given in [3]. In hard exploration games such as PrivateEye, Solaris, and Montezuma's Revenge, both our method and the baselines cannot outperform random policy. Also, in Centipede and TimePilot, our method obtains similar scores to random policy. We will highlight the limitations of our algorithm in our revision.
> > > >
> > > > The results of Montezuma's Revenge are described in Appendix E.6 in our paper. We describe the result in text and do not give the learning curve. We have added the result in [3], showing that it scores zero in the training process for all methods. We give a video of trained policy in [4]. From the video, we find that the self-supervised agent with DB-bonus can learn some skills, including stepping down the ladder, jumping to the pillar, and trying to escape the enemy. Nevertheless, as the learning curve suggests, such learned skills along are insufficient for obtaining scores.
> > > >
> > > > [1] Mnih, Volodymyr, et al. "Human-level control through deep reinforcement learning." nature 518.7540 (2015): 529-533.
> > > >
> > > > [2] Wang, Ziyu, et al. "Dueling network architectures for deep reinforcement learning." ICML 2016
> > > >
> > > > [3] See experiment results with a random-policy baseline at: https://github.com/review-anon/DB
> > > >
> > > > [4] See video of Montezuma's Revenge with DB bonus at: https://www.dropbox.com/s/boijqmt66mgnj17/montezuma-DB.mp4?dl=0

---

### Official Review · Reviewer_1fqw · 2021-07-17

**Rating:** 6
**Confidence:** 4

**Summary:**

This paper proposes a new self-supervised exploration method for RL that encourages an agent to explore states which are more relevant in terms of information gain. To this aim, the Information-Bottleneck (IB) principle has been utilized to learn a model that can obtain dynamics-relevant information and discard dynamics-irrelevant features together. The paper evaluates the proposed technique in multiple Atari environments without accessing the extrinsic rewards.

**Limitations And Societal Impact:**

This paper discusses a new method to encourage an RL agent to explore state-action pairs with high information gain which doesn't have a social impact. As such, I don't think this work has any negative societal impact.

**Main Review:**

Overall, the main idea of the paper is interesting and it is well explained and well written. In addition, the experiments in this paper support the main claim of the paper and show better performance than some of the curiosity-driven methods.

1) While the idea of the paper seems promising, utilizing information-bottleneck for exploration has been well-studied and it seems this paper has a good overlap with previous works such as [1] and [2]. The main difference that I see between this work and previous works is using contrastive learning. Is it fair? or are there many more differences between them?

2) It is not clear to me why I([St, At]; Zt) should be minimized rather than maximized? Since the main claim of the paper is to acquire dynamics-relevant information and discard dynamics-irrelevant, why does minimizing I([St, At]; Zt) not result in discarding "relevant" information? I might miss something here, but can the authors elaborate on this?

3) There'd have been interesting if the authors had included experiments in which reward = DB-Bonus + extrinsic-reward for a game like Montezuma’s Revenge. The reason I mention this is because if combining these two works, it further supports the claim of this paper (i.e. acquiring dynamics-relevant information and discard irrelevant ones).

4) It is not clear how negative samples are collected for Eq 5? What does "sampling observation encodings randomly from the batch" [line 143] mean? Don't we do the same thing for regular samples ( i.e. positive ones)?

5) Is it fair to say that Theory 1 is not that useful and relevant as it only holds without the contrastive objective? as such it shouldn't be claimed as a contribution in this paper.

6) SSE-DB seems only better than 12-13 tasks, not 15 tasks (Not better in Tennis, Crazy Climber, Centipede, Time Plot, MasPAcman). Refer to results in Figure 3. Is that right?

Minor comments:
- Explain the information-bottleneck principle in the paper or at least in the appendix. The paper should be self-contained.

- Explain what is "data processing inequality" in line 222.

- What is \mu in line 191?

- Use a different notation for variational approximation r(z_t), it can be mistaken with reward function.


[1] Curiosity-Bottleneck: Exploration By Distilling Task-Specific Novelty, Youngjin Kim, Wontae Nam, Hyunwoo Kim, Ji-Hoon Kim, Gunhee Kim

 [2] EMI: Exploration with Mutual Information
Hyoungseok Kim, Jaekyeom Kim, Yeonwoo Jeong, Sergey Levine, Hyun Oh Song

**Time Spent Reviewing:**

4

---

> ### Author Response · Authors · 2021-08-10
> **Response to reviewer 1fqw**
>
>
> We thank the reviewer for the valuable comments and time dedicated to evaluating our work.
>
> 1. Our method is different from CB [1] and EMI [2] in the following aspects.
>
> - CB learns to extract task-relevant information, where 'task' is identified by environmental rewards. However, when the environmental rewards are sparse or entirely unavailable (which is studied in our paper), CB will hardly work since it relies on environmental rewards. In our experiments, CB fails in most tasks in self-supervised setting. In contrast, DB enables to perform self-supervised exploration efficiently. Meanwhile, CB uses the compressiveness of observation with respect to the Q-network as the intrinsic reward, which is intuitive but not theoretically grounded. In contrast, we provide theoretical analysis to show that DB-bonus is closely related to the provably efficient UCB-bonus and count-based bonus.
>
> - EMI [2] learns a representation by maximizing the Mutual Information (MI) in the forward dynamics (i.e., $I([S,A; S'])$) and the inverse dynamics (i.e., $I([S,S']; A)$), which is different from the Information Bottleneck (IB) principle used in our method. In addition, we aim to perform robust exploration to overcome the white-noise problem, while EMI does not have an explicit mechanism to address the noise. To verify this, we add additional experiments by using the official EMI code with additional random-box noise in observations. We run experiments in 5 Atari games in a self-supervised exploration setting. The results show EMI performs not good without extrinsic reward. In addition, we find that EMI is not robust to white-noises as well. The modified code and results are released in an anonymized link [3].
>
> 2. Our intention in minimizing $I([S_t, A_t]; Z_t)$ is to compress the representation. In fact, setting $Z_t = [S_t, A_t]$ will ensure that $I([S_t, A_t]; Z_t)$ is maximized. However, such encoding is not desirable as it contains irrelevant noise that might harm exploration. To avoid this problem, we minimize $I([S_t, A_t]; Z_t)$ and consider it as a regularizer in the representation learning. On top of minimizing such MI, the representation learning is done by maximizing the MI $I(Z_t; S_{t+1})$. Maximizing such MI ensures that we do not discard useful information from $(S_t, A_t)$. The entire learning process is motivated by the Information Bottleneck (IB) principle, which aims to extract information through minimizing the MI between representation and input while maximizing the MI between representation and label simultaneously.
>
> 3. Montezuma's Revenge is a hard exploration task with difficulties in searching and planning. Exploration methods that work well on this task often require specific treatments. Taiga et al [1] extensively compared exploration methods on Montezuma's Revenge. The results show that although some methods achieve good performance in Montezuma's Revenge (e.g. RND obtains approximately 8150 points with extrinsic rewards), they do not perform well on other Atari games (e.g, RND performs similar to $\epsilon$-greedy in other Atari games). We add an experiment to train DB in Montezuma's Revenge with extrinsic rewards being available, which achieves around 4120 points. We also provide a visualization analysis in Appendix E.6, which shows that the latent representation in DB is well aligned in several clusters. Although the obtained score may be suboptimal comparing with the best baselines solving Montezuma's Revenge, we have to clarify that our proposed SSE-DB is not specifically designed to solve such a task. Moreover, DB focuses on robust exploration and has more advantages in noisy environments according to the empirical results in Section 5.2 and Appendix E.
>
> 4. The positive samples are obtained by directly sampling the transitions $(s, a, s')$. In contrast, the negative samples are obtained by first sampling a state-action pair $(s, a)$, and then sampling a state $\tilde s$ independently from the batch. Finally, a negative sample point is obtained by concatenating them together to form a tuple $(s, a, \tilde s)$. Note that the tuple in negative samples does not follow the transition dynamics, whereas the positive samples follow the distribution of transition dynamics.
>
> 5. In our work, we present Theorem 1 to show that our bonus is well-motivated and is related to provably efficient bonuses in the study of RL theory. We adopt the predictive loss in our analysis as it is more tractable and allows us to compute the mutual information of representations in closed form. In our proposed algorithm, the contrastive loss is an empirical approach that allows us to estimate the mutual information in the loss function. In our experiments, we find that incorporating a contrastive loss is important and it leads to a stronger distillation for the dynamics-relevant feature. Characterizing the theoretical property of contrastive learning in linear MDP is also an important future direction of our research.
>
> 6. We have 18 tasks in total. Although in the *standard* Atari games, DB outperforms other methods in 15 of 18 tasks, in the *noisy* Atari environments with random-box noise (shown in Appendix E.1), our method outperforms other methods in 17 out of 18 tasks. Our experiments show that SSE-DB is better for environments with dynamics-irrelevant information.
>
> Minor:
>
> 1. Thanks for pointing out this problem. In learning a representation $Z$ of a given input source $X$ with the target source $Y$, IB maximizes the mutual information between $Z$ and $Y$ (i.e. $\max I(Z; Y)$) and restricts the complexity of $Z$ by using the constrain as $I(Z;X)<I_c$. Combining the two terms, the objective of IB is equal to $\max I(Z;Y)-\beta I(Z;X)$ with the introduction of a Lagrange multiplier. In DB, $X$ represents the current state-action, and $Y$ represents the next state. We want to learn a robust representation by following the IB principle. We will add a background section of IB in revision.
>
> 2. Data Processing Inequality (DPI) is an information theoretic concept that can be understood as: 'post-processing' cannot increase information. In line 222, since $g(s_t,a_t,S_{t+1})$ is a post-processing of $(s_t,a_t,S_{t+1})$, we have $I(\Theta; (s_t,a_t,S_{t+1}))>I(\Theta; g(s_t,a_t,S_{t+1}))$, where $g$ is a neural network in practice.
>
> 3. In linear MDP, the feature map of the state-action pair is denoted as $\eta:\mathcal{S}\times\mathcal{A}\rightarrow\mathbb{R}^d$. The transition kernel and reward function are assumed to be linear to $\eta$. $\Lambda_t$ is a Gram matrix that accumulates features of previous samples. Due to space limitation, we did not explain the UCB-bonus in detail in the main text. The details are provided in Appendix C.1. We do not find $\mu$ in line 191.
>
> 4. Thanks for the suggestion. We will replace $r(z_t)$ with a different notation.
>
> References
>
> [1] Curiosity-Bottleneck: Exploration By Distilling Task-Specific Novelty, Youngjin Kim, Wontae Nam, Hyunwoo Kim, Ji-Hoon Kim, Gunhee Kim
>
> [2] EMI: Exploration with Mutual Information Hyoungseok Kim, Jaekyeom Kim, Yeonwoo Jeong, Sergey Levine, Hyun Oh Song
>
> [3] See experiments of EMI with white-noise at: https://github.com/review-anon/EMI-White-Noise
>
> [4] Taiga, Adrien Ali, et al. "On bonus based exploration methods in the arcade learning environment." International Conference on Learning Representations. 2020.

---

> > ### Comment · Reviewer_1fqw · 2021-08-24
> > **Thanks for your responses.**
> >
> > Thanks for your responses.
> > While I still stand by my comment about Th1, I am happy with the rest of responses.

---

### Official Review · Reviewer_L2WS · 2021-07-21

**Rating:** 8
**Confidence:** 4

**Summary:**

In this paper, the authors introduced Dynamic Bottleneck (DB) model that learns dynamics-relevant representations based on the Information Bottleneck principle. They further proposed DB-bonus for efficient exploration and established theoretical connections between the proposed DB-bonus and provably efficient bonuses. The experiments show that the proposed method outperforms several strong baselines in stochastic environments for self-supervised exploration.

**Main Review:**

Pros:
1. The information-based DB method is novel in combating noise in observation.
2. The variational methods help to scale the method up to solve high dimensional problems.
3. The authors show the link of the DB bonus to two provably efficient cases: linear MDP and tabular MDP.
4. The ablation studies are sufficient that unveil interesting patterns latent space.

Cons:
1. I am a little bit confused about the definition of “noisy states”. In the introduction, the authors give an example: “For example, in autonomous driving tasks, the states captured by the camera may contain irrelevant objects, such as clouds, birds, and aircraft.” However, most of these objectives may would have their own dynamics, it is just those dynamics are irrelevant to rewards. Could the proposed method be able to handle disturbance with dynamics? If not, I suggest change this example and make it clear in the paper. If so, then maybe add experiments in this case.

Typos:
1. The “log” operators are missing in Eq. 7.
2. There are some obvious grammar errors.

**Time Spent Reviewing:**

10

---

> ### Author Response · Authors · 2021-08-10
> **Response to reviewer L2WS**
>
> We thank the reviewer for the valuable comments and time dedicated to evaluating our work.
>
> We agree that the motivating example mentioning 'clouds, birds, and aircraft' is somehow inappropriate since some of the objects have their own dynamics. Our intention is to highlight that these random objects captured as part of the pixel input will harm novelty-based exploration. To make our point precise, let us consider a transition without any distractors as $(s, a, s')$, and the transition of an irrelevant and stochastic object (e.g., cloud) as $(c, c')$, which behaves similar to Brownian movement. Then, the complete transition is denoted as $([s, c], a, [s', c'])$. DB is proposed to overcome this kind of stochasticity that comes from the task-irrelevant noise.
>
> To instantiate such noise, we inject various noisy patterns (i.e., random-box, pixel noise, and sticky actions) into the environments. These patterns can not represent all the realistic noises encountered in real-world applications, but can provide meaningful distractors that disturb exploration. How to construct more realistic distractors is an interesting direction worth exploring in the future. We will add more discussion about this problem in our revision.

---

### Decision · Program_Chairs · 2021-09-27

**Decision:**

Accept (Poster)

**Comment:**

Paper presents a method for intrinsic exploration based on learning a representation of observation that can ignore distractors using the dynamics bottleneck principle. Optimality of the exploration bonus in the case of bandits is established. Results on standard benchmarks used in the exploration literature are convincing. Reviewers unanimously vote to accept the paper, which I agree with.